

# Field heterogeneity of soil texture controls leaf water potential spatial distribution in non-irrigated vineyards

Louis Delval[1], Jordan Bates[2], François Jonard[1,3], Mathieu Javaux[1,2]

[1]Earth and Life Institute, Environmental Sciences, UCLouvain, 1348, Louvain-la-Neuve, Belgium

[2]Agrosphere IBG-3, Forschungszentrum Jülich GmbH, 52428, Jülich, Germany

[3]Earth Observation and Ecosystem Modelling Laboratory, ULiège, 4000, Liège, Belgium

*Correspondence to*: Louis Delval (louis.delval@uclouvain.be)

**Abstract.** Grapevine water status exhibits substantial variability even within a single vineyard. Understanding how edaphic, topographic and climatic conditions impact grapevine water status heterogeneity at the field scale, in non-irrigated vineyards, is essential for winemakers as it significantly influences wine quality. This study aimed to quantify the spatial distribution of grapevine leaf water potential ($\Psi_{leaf}$) within vineyards and to assess the influence of soil properties heterogeneity, topography and weather on this intra-field variability, in two non-irrigated vineyards during two viticultural seasons. By combining multilinearly vegetation indices from very-high spatial resolution multispectral, thermal and LiDAR imageries collected with unmanned aerial systems, we efficiently and robustly captured the spatial distribution of $\Psi_{leaf}$ across both vineyards, at different dates. Our results demonstrated that in non-irrigated vineyards, the spatial distribution of $\Psi_{leaf}$ was mainly governed by the within-vineyard soil hydraulic conductivity heterogeneity ($R^2$ up to 0.81), and was particularly marked when the evaporative demand and the soil water deficit increased, since the range of $\Psi_{leaf}$ was greater, up to 0.73 MPa, in these conditions. However, topographic attributes (elevation and slope) were less related to grapevine $\Psi_{leaf}$ variability. These findings show that soil properties within-field spatial distribution and weather conditions are the primary factors governing $\Psi_{leaf}$ heterogeneity observed in non-irrigated vineyards, and their effects are concomitants.





## 1 Introduction

Accurately quantifying grapevine water status is crucial for winemakers as it significantly impacts wine quality (Dry & Loveys,
1999; Van Leeuwen et al., 2009). Detailed spatial information on grapevine water status can be particularly useful for providing
guidelines on viticultural management to optimize grape production. This is especially important in the context of climate
change, which poses crucial challenges for fresh water use in viticulture (Gambetta et al., 2020). Many authors have
demonstrated that grapevine water status exhibits substantial variability even within a single field, particularly when significant
water restriction is found (Acevedo-Opazo et al., 2010; Brillante, Martínez-Luscher, et al., 2017; Tisseyre et al., 2005).
Measuring leaf and/or stem water potential is an effective method to assess grapevine water status (Choné et al., 2001) but
accurate measurements of leaf and stem water potentials are usually achieved on single plants using a Scholander pressure
bomb (Scholander et al., 1965) or with psychrometers. Several authors reported high magnitude of variation of leaf water
potential over the viticultural season (e.g. 1.6 MPa in Ojeda et al. (2005)) and at the within field level (e.g. 1.2 MPa in Ojeda
et al. (2005); 0.7 MPa in Brillante et al. (2017a)). Yet, these methods are time-consuming and labor-intensive and are therefore
not effective to capture instantaneously within-field heterogeneity of grapevine water status (Romero et al., 2018), particularly
under heterogeneous soil and microclimatic conditions generally observed in a vineyard.

Remote sensing technological advances give good opportunities for time- and cost-efficient detection of spatial and temporal
variability of plant water status (Acevedo-Opazo et al., 2008). Particularly, unmanned aerial systems (UAS) are useful tools
to assist precision viticulture thanks to high spatial resolution imagery, allowing differentiation of row and inter-row
information. UAS can transport different sensors to measure and estimate plant traits, e.g., canopy area, biomass, leaf pigment
concentration or grapevine water status, through vegetation indices (VIs) (Baluja et al., 2012; Poblete et al., 2017; Romero et
al., 2018; Serrano et al., 2010; Zarco-Tejada et al., 2013). Most of the sensors used in precision viticulture are multispectral
sensors, allowing to calculate VIs based on the visible (red, green, blue), red-edge and near infrared (NIR) reflectance of plants
(Ferro & Catania, 2023). Several studies found low to moderate correlations between multispectral VIs and grapevine leaf
water potential, with maximum $R^2$ ranging between 0.4 and 0.5 (Baluja et al., 2012; Espinoza et al., 2017; Romero et al., 2018;
Tang et al., 2022). VIs based on NIR and red-edge bands are better correlated with grapevine water potential as they are greatly
affected by leaf structure and chlorophyll content, both being considered as indicators of grapevine water status (Penuelas et
al., 1997; Rapaport et al., 2015). Recently, machine learning models (i.e., random forest) have been applied to combine
multispectral information from grapevines and predict grapevine water potential. These models performed better ($R^2$ around
0.85) than using single multispectral VIs (Poblete et al., 2017; Romero et al., 2018), however, there was a significant loss of
predictive power between calibration and validation ($R^2$ decreased and RMSE increased significantly) (Tang et al., 2022).



Nevertheless, the combination of several VIs to predict grapevine water status is more efficient since each VI can bring complementary information (Xue & Su, 2017).


In dry conditions, grapevine closes stomata, limiting transpiration but preventing it from reaching excessively negative water potentials that could lead to xylem cavitation and death (Gambetta et al., 2020). Once stomata are closed and transpiration is restrained, leaves temperature increases (Costa et al., 2010). Canopy temperature can therefore potentially be used to develop index giving information on stomatal conductance and leaf water potential (H. G. Jones et al., 2002). Thermal VIs such as crop

water stress index (Idso et al., 1981) has shown moderate (but better than multispectral VIs) correlations with grapevine water potential, with a $R^2$ of 0.55 (Romero et al., 2018). Some studies even showed a significant correlation, with $R^2$ around 0.80, between leaf water potential and crop water stress index in Mediterranean vineyards (Bellvert et al., 2014; Möller et al., 2007). Thermal sensors are not as used as multispectral sensors in precision viticulture (Ferro & Catania, 2023). These sensors are generally more expensive and subjected to less straightforward calibration (Berni et al., 2009). Thermal data processing is less

simple than multispectral data but complementary information obtained by multispectral and thermal sensors could improve the ability to remotely monitor grapevine leaf water potential (Tang et al., 2022).

The use of laser scanning sensors (Light Detection and Ranging - LiDAR) for estimating biophysical parameters of vineyard canopy is still relatively uncommon compared to other available sensors (Ferro & Catania, 2023). Point clouds obtained from

LiDAR sensors are suitable for detecting structural features of grapevine such as canopy height, canopy width or even leaf area index (Bates et al., 2021; Comba et al., 2018). These structural features reflect the result of cumulative water potential of grapevine, and could therefore contain information on grapevine water status (Baluja et al., 2012). As a result, the combination of multispectral, thermal and LiDAR (structural) data obtained from different sensors onboard unmanned aerial vehicles could potentially be used to improve the mapping of grapevine leaf water potential within a vineyard.


In addition to the value of remotely monitoring grapevine leaf water potential in an accurate way through UAS, it is also interesting to study which factors spatially determine leaf water potential at the vineyard scale. Several studies try to assess the influence of irrigation management on grapevine water status with UAS platforms (De Bei et al., 2011; Bellvert et al., 2012, 2015; Espinoza et al., 2017; Möller et al., 2007). However, most of these studies were conducted on vineyards with

homogeneous edaphic conditions, but different irrigation treatments. It remains unclear how robust these water status mapping approaches are when applied across vineyards with varying edaphic and meteorological conditions (Helman et al., 2018). Few studies evaluated how spatialized information of grapevine water status give us information about how environmental heterogeneity within a vineyard affects the distribution of grapevine water potential (Brillante, Martínez-Luscher, et al., 2017). It is known that the soil and the climate mainly affect grapevine water potential, and their effects are concomitant (Van

Leeuwen et al., 2004). The soil, through its texture and its ability to retain and conduct water, determines the water supply to the root system. Soil depth, texture, structure and percentage of coarse elements, also affect the growth of the root system and



therefore the available water for the plant (Van Leeuwen et al., 2018). Grapevine water status is also affected by the vapor pressure deficit (VPD, corresponding to the atmospheric evaporative demand), which depends on temperature and air humidity (Soar et al., 2006). Grapevine should therefore find an equilibrium between water supply, in the soil, and water demand, in the

atmosphere, by regulating its water potential and stomatal conductance, to maintain gas exchanges for photosynthesis, while preventing excessive negative water potential leading to xylem cavitation (Gambetta et al., 2020). Topographic attributes, such as slope and elevation, can also impact grapevine performance (Bramley et al., 2011; Karn et al., 2024). On the one hand, this influence is indirect since topography control the redistribution of soil particles within the vineyard and therefore creates soil texture spatial heterogeneity (Fraga et al., 2014). On the other hand, topography can directly influence grapevine water status

through its impact on water drainage and runoff, sunlight exposure and temperature variations within the vineyard (Brillante, Martínez-Luscher, et al., 2017; Karn et al., 2024). These effects foster microenvironments within a vineyard that could affect grapevine water status (Rabia et al., 2022).

In this study, we aimed to quantify the spatial distribution of grapevine leaf water potential within a vineyard and to assess the

impact of edaphic, topographic and climatic conditions on this intra-field heterogeneity. We evaluated the capabilities of UAS equipped with multispectral, thermal and LiDAR sensors to monitor grapevine leaf water potential, on two non-irrigated vineyards, during two viticultural seasons.

## 2 Methodology


### 2.1 Site descriptions

This study was conducted on two non-irrigated Belgian vineyards, grassed in the inter-rows, namely the *Château de Bousval* vineyard and the *Domaine W* vineyard (Fig.1.a). At the *Château de Bousval* vineyard (Genappe, Belgium, 50°36'45.0''N

4°31'19.6''E), we focused on an east-facing plot of Chardonnay grafted on 3309C rootstock, planted in 2014 with vertical shoot positioning, 1.6 m inter-row and 0.8 m inter-cep. The field rises between 110 m and 125 m above sea level (Fig.1.b), and the average slope is 6 %. At the *Domaine W* vineyard (Tubize, Belgium, 50°41'19.4"N 4°09'36.9"E), two Chardonnay plain fields grafted on 101-14Mgt rootstock were selected for this study, rising between 52 m and 54 m above sea level (Fig.1.g), with rows oriented north-south. The grapevines were planted in 2016 with vertical shoot positioning, a 2.2 m inter-

row and 1 m inter-cep.





In the *Château de Bousval* vineyard, the soil is made of a loamy top layer overlying a sandy subsoil, but the depth of the interface between these two layers changes within the plot, reaching more than 3 m at the lowermost side (east side) of the field, due to an accumulation of loamy colluviums. At the upper part (west side) of the field, the loamy layer is around 0.4 m

depth. We know, on the whole field, the depth of the interface between the interface between the loam and the sand (Fig.1.c). Moreover, in this vineyard, we also know that grapevine roots reach a depth of at least 2.5 m on the whole field (Delval et al., submitted). In the *Domaine W* vineyard, the soil heterogeneity is less marked in terms of soil texture. The north-western part of the field is defined by a silty loam soil on the first horizons of the profile and silty clay loam soil thereafter. The south-eastern part is defined by a silty loam soil on the whole profile (Fig.1.h). These differences in terms of soil texture have only

been observed in a single location in each subplot. Therefore, unlike Bousval, the depth of the interface between silty loam soil and silty clay loam soil is not accurately known on the whole field. A stream runs adjacent to the southeast parcel, raising the water table in this area within the reach of the roots. Root depth on this vineyard is at least 2 m everywhere in the field (Delval et al., submitted).

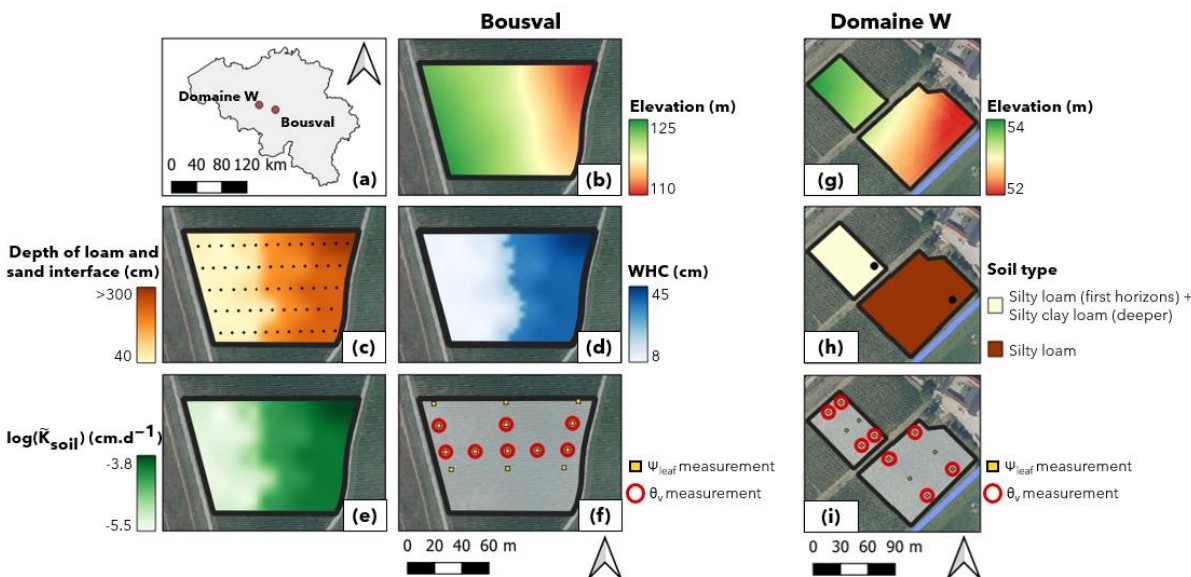

**Figure 1 – (a)** Location of the vineyards in Belgium. **(b-f)** Topographic attributes, edaphic properties and location of measurements in the *Château de Bousval* vineyard: **(b)** Elevation; **(c)** Kriged map of the depth of the interface between the loamy soil and the sandy subsoil (the black points are locations of soil samples used to determine the depth of the interface *in situ* and used for kriging); **(d)** Water holding capacity (WHC) on 2.5 m depth (equation 6); **(e)** Averaged soil hydraulic conductivity ($\widetilde{K}_{soil}$) on 2.5 m depth (equation 7); **(f)** Locations of the leaf water potential ($\Psi_{leaf\_meas}$) and volumetric water

content ($\theta_v$) profile measurements. **(g-i)** Topographic attributes, edaphic properties and location of measurements in the *Domaine W* vineyard: **(g)** Elevation; **(h)** Soil type (the black points are locations of soil samples to determine soil type *in situ*);





**(i)** Locations of the leaf water potential ($\Psi_{leaf\_meas}$) and volumetric water content ($\theta_v$) profile measurements. In **(g)**, **(h)** and **(i)**, the blue line is a stream adjacent to the south-eastern plot.

**2.2 Meteorological conditions**

Both vineyards are equipped with weather stations providing hourly meteorological data. The meteorological conditions during each flight are provided in Table 1. The atmospheric conditions and the evaporative demand are characterized with the vapor pressure deficit (VPD). The daily water deficit, which refers to the standardized precipitation evapotranspiration index (SPEI

- Vicente-Serrano et al., 2010), is calculated as follows:

$$SPEI = \sum_i \left(P_i - ET_{0_i}\right), \tag{1}$$

with i = 1 corresponds to the 1st April of the respective year (budburst of grapevines), $P_i$ the daily precipitation, and $ET_{0i}$ the

daily reference evapotranspiration. The more negative the SPEI, the greater the water deficit. $ET_0$ is calculated from the FAO Penman-Monteith method (R. Allen et al., 1998).

**Table 1** – Meteorological conditions during each flight campaign.

| | Bousval | | | Domaine W | | |
|---|---|---|---|---|---|---|
| **Date** | *Air temperature (°C)* | *VPD (kPa)* | *SPEI (mm)* | *Air temperature (°C)* | *VPD (kPa)* | *SPEI (mm)* |
| 27/07/22 | 18.9 | 1.02 | -273.9 | | | |
| 10/08/22 | 28.3 | 1.95 | -334.6 | 28.9 | 1.89 | -276.2 |
| 31/08/22 | 22.5 | 1.15 | -411.3 | 23 | 1.32 | -340.2 |
| 20/07/23 | 20.9 | 1.02 | -193.0 | 21 | 1.2 | -193.8 |
| 10/08/23 | 23.1 | 1.53 | -197.4 | | | |
| 06/09/23 | 27.9 | 2.17 | -223.6 | 29.9 | 2.08 | -236.9 |






Despite relatively similar air temperatures and VPDs during the 2022 and 2023 flights, 2022 was a drier year than 2023. The SPEI for any date in 2022 is significantly more negative than that for any date in 2023 in both vineyards (Table 1), indicating drier conditions in 2022 than 2023.

**2.3 Data acquisition and processing**

Unmanned aircraft systems (UAS) data acquisition took place during the vine growth period in 2022 and 2023. Three flight campaigns were carried out in 2022 and three in 2023, for a total of six flight campaigns. Data were acquired before the veraison (27/07/22 and 20/07/23), at the start of the veraison (10/08/22 and 10/08/23) and just before harvest (31/08/22 and

06/09/23). Flights started around 12:00 (UTC +2) at Bousval and around 13:30 (UTC +2) at Domaine W. Three sensors were used in this study, a Micasense RedEdge-M multispectral sensor (Micasense Inc., Seattle, WA, USA), a FLIR Vue Pro R thermal camera (FLIR Systems, Wilsonville, OR, USA) and a YellowScan Surveyor LiDAR (YellowScan, Saint-Clément-De-Rivière, France), mounted on a DJI Matrice 600 (SZ DJI Technology Co Ltd., Shenzen, China). The Micasense RedEdge-M is a five narrowband multispectral camera, capturing blue (465-485 nm), green (550-570 nm), red (663-673 nm), red-edge

(712-722 nm) and NIR (820-860 nm) wavelengths of the electromagnetic spectrum. Just before and after each flight, the Micasense RedEdge-M sensor was calibrated thanks to a reflectance panel to ensure accurate and consistent reflectance measurements, enabling reliable comparisons of data captured under varying light conditions and at different times. Images were acquired to ensure approximately 90 % forward and lateral overlap. To process the multispectral imagery, orthomosaics were first created for each band of the sensor, using Pix4D (Pix4D, Lausanne, Switzerland). Ground control points (GCP) were

used for georeferencing.

The FLIR Vue Pro R is a radiometric thermal sensor that captured longwave infrared radiation in the 7.5-13.5 µm range. This sensor needs radiometric calibration parameters such as emissivity of the canopy, air temperature and humidity to capture accurately the surface temperature. Thermal imagery was also processed using Pix4D. The same GCPs as for multispectral imagery were used, enabling georeferencing consistent with multispectral data.


The LiDAR system, operating with a wavelength of 903 nm, is composed of a Velodyne LiDAR puck, onboard computer, Inertial Measuring Units (IMU), and Global Navigation Satellite System (GNSS) receiver. Ranging data are provided by the LiDAR puck. The IMU measured the variations in attitude and orientation, and the GNSS provided positioning. To process

the LiDAR data, YellowScan's CloudStation software was used to align the flight strips for georeferencing and to apply corrections through GNSS offset (lever-arms), sensor-angle (boresight), and GNSS post-processing with precise position techniques (Bates et al., 2021).



Multispectral and thermal sensors were mounted on the same DJI Matrice 600 and data were collected at the exact same time.
The LiDAR data were collected directly after the multispectral and thermal data. Multispectral and thermal flights were
conducted at an altitude of 100 m above ground level and at a flight speed of 6 m.s⁻¹, yielding flight durations of approximately
7 minutes, and resulting in a native pixel resolution of 7 cm for multispectral and thermal data. For the LiDAR flight, the UAS
maintained an altitude of 50 m aboveground, with a ground speed of 5 m.s⁻¹, giving a spatial resolution of 16 cm for LiDAR
data. Due to technical issues, there is no thermal data at Bousval the 27/07/22 and 31/08/22, no multispectral data at Domaine
W the 10/08/22, and no multispectral, thermal and LiDAR data at Domaine W the 27/07/22 and 10/08/23 (Table 2).

**Table 2** – Availability of the UAS data during the six flight campaigns in the two vineyards.

|  | **Bousval** | | | **Domaine W** | | |
|---|---|---|---|---|---|---|
| **Date** | *Multispectral* | *Thermal* | *LiDAR* | *Multispectral* | *Thermal* | *LiDAR* |
| 27/07/22 | Yes | No | Yes | No | No | No |
| 10/08/22 | Yes | Yes | Yes | No | Yes | Yes |
| 31/08/22 | Yes | No | Yes | Yes | Yes | Yes |
| 20/07/23 | Yes | Yes | Yes | Yes | Yes | Yes |
| 10/08/23 | Yes | Yes | Yes | No | No | No |
| 06/09/23 | Yes | Yes | Yes | Yes | Yes | Yes |


At the same time as collecting UAS data, we measured grapevine leaf water potential ($\Psi_{leaf\_meas}$) on various 2x2 m² zones
homogeneously distributed across the fields, using a Scholander pressure bomb (670 Pressure Chamber, PMS Instrument
Company). The $\Psi_{leaf\_meas}$ was measured on 14 2x2 m²zones at Bousval (Fig.1.f), and on twelve 2x2 m² zones at Domaine W
(Fig.1.i). For each sampled grapevine, $\Psi_{leaf\_meas}$ were recorded on three to five mature leaves, covered by an aluminium zip
bag 45 minutes before the measurement. In addition, the soil water content profile to a depth of 105 cm (every 15 cm) was
also measured in eight measurement zones in each vineyard (Fig.1.f and Fig.1.i) , before each flight, with a TRIME-FM3 time
domain reflectometry combined with access T3 tube (IMKO GmbH, Ettlingen, Germany).






**2.4 Pure grapevine pixels extraction**

The fine spatial resolution of the UAS data makes it possible to distinguish rows and inter-rows. We used the k-means algorithm to generate a binary mask that distinguishes pure vine canopy pixels from inter-row soil and grass pixels. The k-means algorithm determined the optimal thresholding value to maximize the between-class variance and minimize the within-class variance (MacQueen, 1967). K-means algorithm has already shown good ability to extract pure vine pixels based on multispectral vegetation indices (Cinat et al., 2019). This segmentation enables the computation of vegetation indices on grapevine only and facilitates the derivation of $\Psi_{leaf}$ prediction model specifically focused on the grapevine itself. The workflow of the segmentation method used in this study is illustrated in Fig.2. The algorithm was performed using RStudio (RStudio Team, 2022). The method consists of six steps, and was performed for each date, generating a mask per date:

1) The initial step was to use the canopy height, derived from the LiDAR data, to get a first raw distinction between rows and inter-rows. We derived the height of the canopy (canopy height model CHM) thanks to the difference between the Digital Surface Model (DSM) and the Digital Terrain Model (DTM): CHM = DSM - DTM. We considered that grapevine rows are represented by every pixel greater than 1 m. This step allowed to remove a large part of pixels representing inter-rows.

2) Due to a coarser spatial resolution of LiDAR data, we used multispectral data to get a finer separation and be sure that we extracted only pure grapevine pixels. We identified the most relevant spectral bands capable of distinguishing between vineyard rows and inter-row vegetation to choose a proper vegetation index for the mask creation. The Modified Soil Adjusted Vegetation Index (MSAVI), suggested by Qi et al. (1994), was selected due to its incorporation of the red and NIR bands known for their sensitivity to vegetation density, and its ability to minimise soil brightness influences in sparse crops (Binte Mostafiz et al., 2021). We applied the mask obtained in step 1 on the MSAVI raster.

3) We subdivided the new MSAVI raster into smaller areas (rectangles of 10 m x 1.5 m). By reducing the area in which the k-means algorithm was applied, the non-vine pixels in that area exhibited greater similarity, allowing the algorithm to better discriminate between grapevine and grass pixels.

4) The k-means algorithm was then applied in each rectangle. The number of clusters was set to three to distinguish among the three classes identifiable within the rectangles: pure grapevine pixels, mixed pixels, pure grass pixels.

5) Following the algorithm execution, only the class with the highest mean value of MSAVI was retained, aiming to automatically extract grapevine class. This class was expected to have the highest MSAVI value since it represents the class with highest biomass density.

6) Finally, the outputs of the algorithm were combined to create a unified shapefile, representing a binary mask that isolates pure grapevine canopy pixels across the field. This mask was subsequently utilized to filter out non-vine or mixed pixels from UAS data, and only the remaining grapevine pixels were used for further analysis.





245

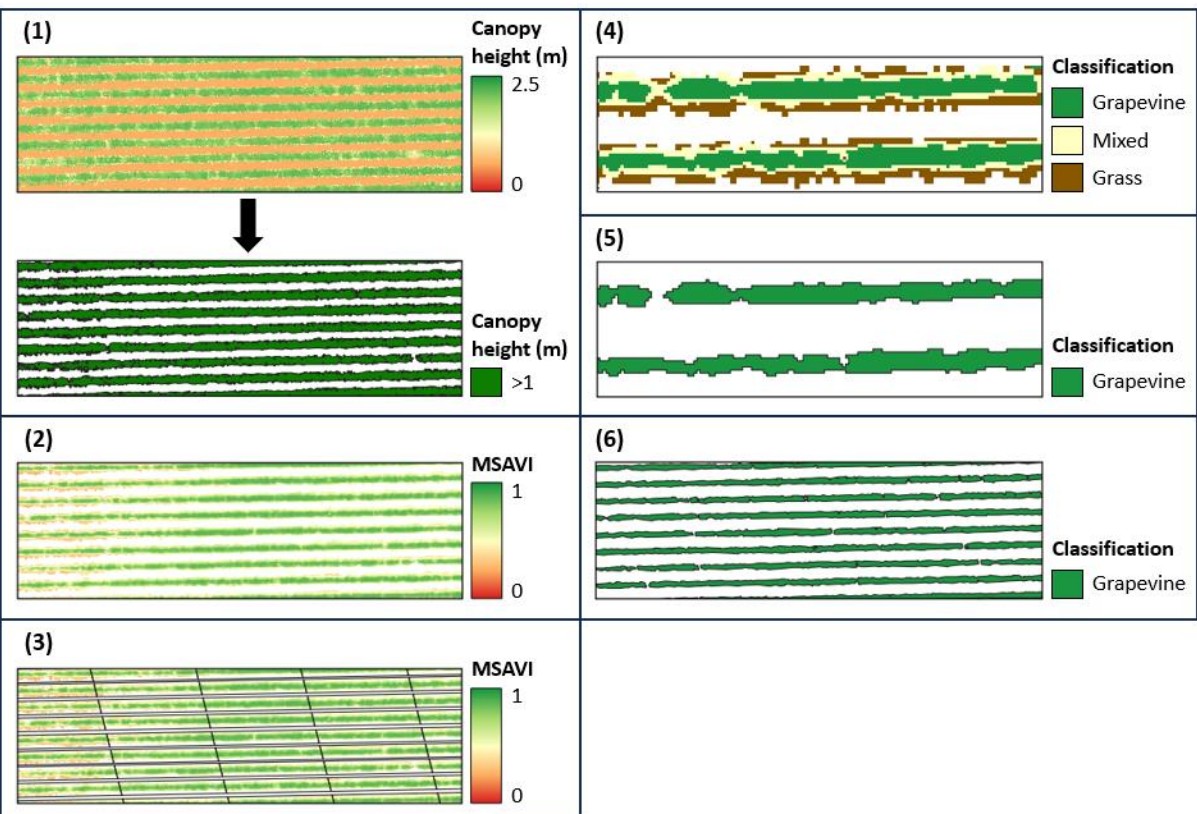

**Figure 2** – Process of segmenting grapevine pixels from inter-row pixels to generate a pure grapevine mask. **(1)** First raw distinction between rows (>1 m) and inter-rows (<1 m) using the canopy height derived from the LiDAR data; **(2)** Application of the raw grapevine mask obtained in (1) on the Modified Soil Adjusted Vegetation Index (MSAVI) raster; **(3)** Subdivision of the new MSAVI raster obtained in (2) into smaller areas (rectangles of 10 m x 1.5 m); **(4)** Application of k-means algorithm in each rectangle. The number of clusters was set to three (pure grapevine, mixed, pure grass); **(5)** Selection of pure grapevine pixels, based on the clustering obtained with the k-means algorithm; **(6)** Creation of a unique mask to extract pure grapevine pixels from each raster of vegetation indices.


**2.5 Generation and extraction of UAS-based variables**

We calculated multiple widely used multispectral vegetation indices (VIs), exploiting different band combinations. The multispectral VIs used in this study were obtained from the review of Giovos et al. (2021). 43 multispectral VIs were retained





in this study, as they constitute the most frequently used to monitor and estimate vine water stress and delineation of management zones in viticulture. We also exploited the specific red, blue, green, NIR and red-edge bands alone. The complete list of all the indices and how they are calculated can be found in Table S1.

In addition, we calculated thermal VIs, namely the canopy surface temperature (CST [°C]) and the difference between CST
and air temperature (dT = CST - Ta [°C]). The Crop Water Stress Index (CWSI) was also computed using the simplified formula suggested by Jones (1992):

$$CWSI = \frac{CST - T_{wet}}{T_{dry} - T_{wet}}. \tag{2}$$

Two different CWSI were derived:

(a)   CSWIa: $T_{dry}$ and $T_{wet}$ were measured during the UAS surveys by a meteorological station.
(b)   CWSIb: $T_{dry}$ and $T_{wet}$ were derived from the pure canopy pixels and were respectively considered as $CST_{max}$ and $CST_{min}$.

Structural features of the grapevines were derived from the LiDAR data. We derived the height of the canopy (CH) thanks to the difference between the DSM and the DTM as explained before. We also derived the leaf area index (LAI) with the LiDAR
method developed by Bates et al. (2021). From the LiDAR data, we also derived the elevation and the slopes of the different vineyards, but these parameters were not used to predict $\Psi_{leaf}$ since they do not vary over time. However, they were used to interpret the spatial distribution of $\Psi_{leaf}$.

After applying the binary mask to isolate pure vine pixels on each index map described above, we extracted the averages of
the index values within the 2x2 m² zones defined before for the grapevine leaf water potential measurements (Fig.1.f and Fig.1.i).

## 2.6 Leaf water potential prediction

As a preliminary step, we examined the univariable relationships between *in situ* $\Psi_{leaf\_meas}$ and the different VIs, based on Pearson's coefficient (Pearson's ρ), to account for linear relations, and based on Spearman's coefficient (Spearman's ρ), to account for monotonic relations. Person's ρ quantifies the strength and direction of a linear relationship, while Spearman's ρ is valuable for detecting and quantifying associations when non-linear relationships are assumed. We therefore compared Pearson's ρ and Spearman's ρ to evaluate if the relation between $\Psi_{leaf\_meas}$ and a VI was linear or not. These analyses enable
us to assess the capability of simple remotely sensed VIs to evaluate $\Psi_{leaf}$. All in-situ measurement data points (n=132; 6x14 at Bousval and 4x12 at Domaine W) across all vineyards and days, were used for a comprehensive analysis.





We then used the stepwise regression method to develop a multiple linear regression model to predict leaf water potentials ($\Psi_{leaf\_pred}$) based on a multiple linear combination of the VIs presented previously (equation 3). We also predicted an

uncertainty, quantified by the confidence interval of 95 % on $\Psi_{leaf\_pred}$ ($IC_{0.95}{}^{\Psi leaf}$). Stepwise regression is a step-by-step iterative construction of a linear regression model that involves the selection of independent variables to be used in a final model (Wilkinson, 1979). Stepwise regression can be achieved either by trying out one independent variable at a time and including it in the regression model if it is statistically significant (forward selection), or by including all independent variables in the model and eliminating those that are not statistically significant (backward elimination). A combination of both methods is

also possible and was used in this study (bidirectional elimination). This method used the Akaike Information Criterion to add or remove VIs from the multiple linear regression model, minimizing the number of predictor variables but keeping a high predictive power (Akaike, 1974). Stepwise regression models were implemented in RStudio (RStudio Team, 2022), using respectively the *stats* package.


$$\Psi_{leaf\_pred} = \beta_1 \times VI_1 + \beta_2 \times VI_2 + \cdots + \beta_n \times VI_n + \alpha, \tag{3}$$

with $\beta_i$ the regression weights (or beta coefficients) and $\alpha$ the intercept. $\beta_i$ can be interpreted as the average effect on the

predicted variable ($\Psi_{leaf\_pred}$) of a one unit increase in $VI_i$, holding all other predictors fixed. We applied this method on different combinations of VIs and single bands to predict $\Psi_{leaf}$. We tested a total of seven data combinations: (1) multispectral only (M), (2) thermal only (T), (3) LiDAR only (L), (4) multispectral and thermal (M+T), (5) multispectral and LiDAR (M+L), (6) thermal and LiDAR (T+L), and (7) multispectral, thermal and LiDAR (M+T+L).

For each date, we randomly selected 70 % of the data to train the different models and 30 % to validate them. The data used for calibration and validation were the same for each model. We evaluated the performance of the models thanks to the coefficient of determination ($R^2$) and the root mean squared error (RMSE). The model with the best performance was then employed to predict the $\Psi_{leaf}$ over the fields for each flight.

We then verified the reliability of the model. To evaluate that there is no redundancy and similar information in the multiple linear regression model, we used the Variance Inflation Factor (VIF). The VIF assesses if a predictor variable is collinear with the other predictor variables (multicollinearity) in the multiple linear regression model, which could degrade the precision of an estimate, and reduce the reliability and the robustness of the model (P. Allen, 1997). VIF less than 5 indicates a low correlation between a predictor variable and the other ones, VIF between 5 and 10 indicates a moderate correlation, and VIF

greater than 10 indicates a high correlation (James et al., 2021).VIF was computed using the *car* package in RStudio. We also





used partial regression plots to show the effect of a predictor variable on the prediction of $\Psi_{leaf}$, after considering the effects of the other predictor variables. If the slope of the linear regression in a partial regression plot is significantly different from 0 (p-value < 0.001), then it justifies the presence of a predictor variable in the multiple linear model (Moya-Laraño & Corcobado, 2008). The correlation ($R^2$) of the linear model in a partial regression plot allows to quantify the unique relationship between

the predicted variable ($\Psi_{leaf}$) and a predictor variable (VI) while controlling the effects of the other variables. The higher the $R^2$, the greater the influence of the predictive factor (VI) on the predicted variable ($\Psi_{leaf}$) (Zhou et al., 2008).

We carried out unpaired Wilcoxon tests to statistically compare if the median of the $\Psi_{leaf\_pred}$ are significantly different (p-value < 0.05) or not (p-value > 0.05) between dates and vineyards. We performed the Wilcoxon test using the *stats* package

(v4.1.1) of the R Statistical Software (v4.0.4) (RStudio Team, 2022).

**2.7 Relations between leaf water potential and environmental factors**

To understand how environmental conditions influence spatial distribution of $\Psi_{leaf\_pred}$, we analyzed the relations between the

linear combination of VPD and SPEI (Table 1), and the median $\Psi_{leaf\_pred}$ ($\Psi_{leaf\_pred\_median}$) (equation 4), and between the linear combination of VPD and SPEI, and the distribution of $\Psi_{leaf\_pred}$ ($\Psi_{leaf\_pred\_max}$ - $\Psi_{leaf\_pred\_min}$) (equation 5).

$$\Psi_{leaf\_pred\_median} = a \times SPEI + b \times VPD + c, \tag{4}$$


$$(\Psi_{leaf\_pred\_max} - \Psi_{leaf\_pred\_min}) = a \times SPEI + b \times VPD + c, \tag{5}$$

with a and b the regression coefficients, and c the intercept. We applied analysis of covariance (ANCOVA) on to assess if these relations are vineyard-specific or not. Relations were considered statistically different for p-values less than 0.05. For each date, we also quantified the correlation between $\Psi_{leaf\_pred}$ and the elevation, and $\Psi_{leaf\_pred}$ and the slope. At Bousval, since

we accurately know the depth of interface between the loamy soil and the sandy subsoil (Fig.1.c), we also quantified the correlation between $\Psi_{leaf}$ and and the water holding capacity (WHC), and between $\Psi_{leaf}$ and an averaged soil hydraulic conductivity ($\widetilde{K}_{soil}$). We used the coefficient of determination $R^2$ as we assumed that these relations are linear. For these relations, we assumed that grapevines have a uniform root depth of 2.5 m throughout the vineyard. We also assumed, in this study, that the soil unsaturated hydraulic properties of the loamy and sandy soils are the same everywhere in the field. The soil

hydraulic properties were measured by Hyprop (METER Group Inc., Pullman, WA, USA) evaporation method (Bezerra-Coelho et al., 2018). The soil water content at the permanent wilting point (pF 4.2) was measured by pressure plate (Ridley & Burland, 1993). Hyprop-fit software was used to optimize the unsaturated hydraulic parameters of van Genuchten-Mualem





(1980), for each soil texture (Table 3). Everywhere in the Bousval vineyard, we calculated the WHC [cm] (Fig.1.d) and $K_{soil}$ [cm.d$^{-1}$] (Fig.1.e) for a depth of 2.5 m thanks to the following equations:


$$WHC = \left[z_{LS} * \left(\theta(h_{FC}) - \theta(h_{PWP})\right)\right]_{loam} + \left[(250 - z_{LS}) * \left(\theta(h_{FC}) - \theta(h_{PWP})\right)\right]_{sand} \tag{6}$$

with $z_{LS}$ [cm] the depth of the interface between the loamy soil and the sandy subsoil, $\theta(h)$ the water content [cm$^3$.cm$^{-3}$] at the suction h [cm] of the respective soil texture, $h_{FC}$ the suction at the field capacity (in this study we assumed that $h_{FC}$ = -300 cm)

and $h_{FC}$ the suction at the permanent wilting point (in this study we assumed that $h_{PWP}$ = -15000 cm).

$$\widetilde{K}_{soil} = \left[\frac{z_{LS}}{250} * \frac{\int_{h_{FC}}^{h_{PWP}} K(h)dh}{h_{FC} - h_{PWP}}\right]_{loam} + \left[\frac{(250 - z_{LS})}{250} * \frac{\int_{h_{FC}}^{h_{PWP}} K(h)dh}{h_{FC} - h_{PWP}}\right]_{sand} \tag{7}$$

with K(h) the soil hydraulic conductivity [cm.d$^{-1}$] at the suction h [cm].


**Table 3** – Unimodal van Genuchten hydraulic parameters of the loamy soil and the sandy subsoil at the *Château de Bousval* vineyard.

| | $\theta_{sat}$ | $\theta_{res}$ | n | α | $K_{sat}$ | τ |
|---|---|---|---|---|---|---|
| | $cm^3.cm^{-3}$ | $cm^3.cm^{-3}$ | - | $cm^{-1}$ | $cm.day^{-1}$ | - |
| **Loam** | 0.419 | 0.117 | 1.483 | 0.00669 | 1.12 | 0.701 |
| **Sand** | 0.358 | 0.08 | 3.11 | 0.0182 | 72 | 1.039 |

## 3 Results


### 3.1 *In situ* measurements of leaf water potential and relations with UAS-based VIs

$\Psi_{leaf}$ was measured ($\Psi_{leaf\_meas}$) at the same time as the collection of UAS data, at different dates in 2022 and 2023, with a Scholander pressure bomb (Fig.3). We measured a lower median leaf water potential ($\Psi_{leaf\_meas\_median}$ more negative) in 2022

than in 2023 in both vineyards, linked to warmer and drier conditions (table 1). Moreover, in each vineyard, the soil was significantly drier in 2022 than in 2023 (Fig.S1). For a same date, we also measured a slightly lower $\Psi_{leaf\_meas\_median}$ at Bousval



compared to Domaine W, except on 20/07/23 (table 4). We observed a greater $\Psi_{leaf\_meas}$ heterogeneity ($\Psi_{leaf\_meas\_max}$ - $\Psi_{leaf\_meas\_min}$) at Bousval compared to Domaine W. This heterogeneity is even more marked in 2022, when conditions were hot and dry. At Bousval, at each date, the minimum $\Psi_{leaf\_meas}$ was measured at the upper part (west side) of the parcel, where the

loamy soil is shallower, while the maximum $\Psi_{leaf\_meas}$ were measured at the lowermost side (east side) (Fig.3.a), where the loamy soil is deeper. At Domaine W, there is less $\Psi_{leaf\_meas}$ heterogeneity, but we generally measured a lower $\Psi_{leaf\_meas}$ in the north-western plot than in the south-eastern plot (Fig.3.b).


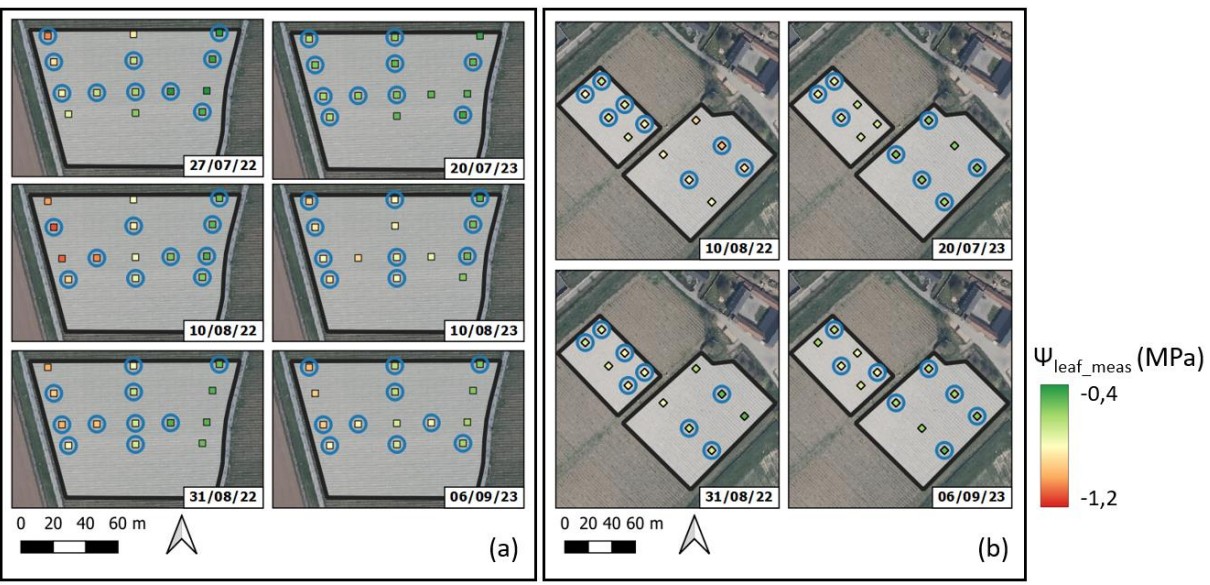

**Figure 3** – Leaf water potential measured ($\Psi_{leaf\_meas}$) with a Scholander pressure bomb in **(a)** 14 zones at the *Château de Bousval* vineyard and **(b)** 12 zones at the *Domaine W* vineyard, at the same time of UAS flights. Points surrounded in blue have been used for the calibration of the multiple linear regression models.






**Table 4** – Median leaf water potential ($\Psi_{leaf\_meas\_median}$), maximum ($\Psi_{leaf\_meas\_max}$) and minimum ($\Psi_{leaf\_meas\_min}$) leaf water potentials (all in MPa) measured in the two vineyards during each UAS flight.

| | Bousval (n=14 by date) | | | Domaine W (n=12 by date) | | |
|---|---|---|---|---|---|---|
| | $\Psi_{leaf\_meas\_median}$ | $\Psi_{leaf\_meas\_max}$ | $\Psi_{leaf\_meas\_min}$ | $\Psi_{leaf\_meas\_median}$ | $\Psi_{leaf\_meas\_max}$ | $\Psi_{leaf\_meas\_min}$ |
| **27/07/22** | -0.73 | -0.45 | -1.00 | | | |
| **10/08/22** | -0.82 | -0.48 | -1.15 | -0.80 | -0.70 | -0.95 |
| **31/08/22** | -0.74 | -0.49 | -1 | -0.71 | -0.55 | -0.83 |
| **20/07/23** | -0.54 | -0.42 | -0.60 | -0.63 | -0.53 | -0.70 |
| **10/08/23** | -0.69 | -0.50 | -0.9 | | | |
| **06/09/23** | -0.72 | -0.49 | -0.95 | -0.70 | -0.55 | -0.90 |

By examining the univariable relationships between $\Psi_{leaf\_meas}$ and the different VIs, including both vineyards and all dates (Fig.S2), we found that the correlation (Pearson's $\rho$) between $\Psi_{leaf\_meas}$ and VIs are low to moderate, ranging from $\rho$ = -0.61 to $\rho$ = 0.63 (Fig.4). We found a maximum $\rho$ of 0.63 between $\Psi_{leaf\_meas}$ and Chlorophyll Red-Edge (CLRededge). The second-best $\rho$ was found with Normalized Difference Red Edge Index NDRE ($\rho$ = 0.62). These two VIs are the only ones, considered in this study, to contain both NIR and red-edge bands, which have been shown to be strongly correlated with the chlorophyll

content of grapevines and therefore influenced by the water status (Laroche-Pinel, Duthoit, et al., 2021; Tang et al., 2022). However, in our case, these $\rho$ values indicate a moderate correlation of these VIs with $\Psi_{leaf\_meas,}$. The third best correlation was found with the thermal index CWSIb, with a Pearson's $\rho$ of -0.61. All others VIs have a lower correlation with $\Psi_{leaf\_meas}$, with $\rho$ ranging between 0.55 and -0.59. This suggests that single VIs cannot accurately predict $\Psi_{leaf}$ in the investigated vineyards and more complex approaches, such as multiple linear regression models, are needed to better estimate $\Psi_{leaf}$. It is interesting

to note that for all indices, the Pearson's $\rho$ (linear relation – Fig.4) is larger than the Spearman's $\rho$ (non-linear relation – Fig.S3). Linear relations between $\Psi_{leaf\_meas}$ and VI have therefore a higher predictive power than non-linear relations (Rebekić et al., 2015), justifying the use of partial linear regression (and not non-linear), to construct a multiple linear regression and predict $\Psi_{leaf}$.







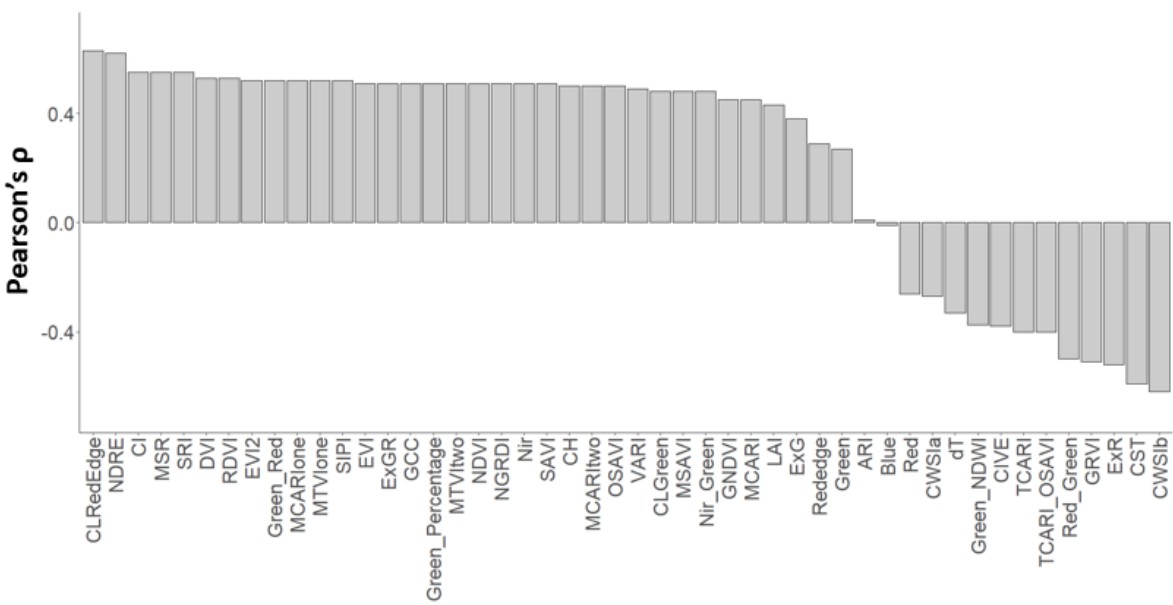

**Figure 4 –** Pearson's coefficient (Pearson's ρ) quantifying the linear correlation between measured $\Psi_{leaf}$ ($\Psi_{leaf\_meas}$) and each vegetation index (VI), by taking all the measurements in both vineyards and at all dates.

**3.2 Predicting leaf water potential based on multiple linear regression model**

In response to the limited correlations obtained from simple linear regressions between $\Psi_{leaf\_meas}$ and VIs, we explored multiple linear regression models to better predict $\Psi_{leaf}$ (Fig.S4). We used the stepwise regression method to minimize the number of VIs in the multiple regression, and to keep the most significative to predict $\Psi_{leaf}$. Figures 5.a and 5.b show respectively the $R^2$

and RMSE obtained by comparing measured $\Psi_{leaf}$ ($\Psi_{leaf\_meas}$) and predicted $\Psi_{leaf}$ ($\Psi_{leaf\_pred}$), for the datasets used for calibration (70 % of the data) and validation (30 % of the data), and for the seven data combinations (see *Methodology* section). The models showed a high consistency between the calibration and validation datasets, with small differences in terms of $R^2$ and RMSE. This showed a great robustness of these models. For example, the model using multispectral, thermal and LiDAR data to predict $\Psi_{leaf\_pred}$ (Fig.5.c) has a $R^2 = 0.80$ and $R^2 = 0.78$ for the calibration and validation respectively, and a RMSE = 0.07

MPa and RMSE = 0.08 MPa for the same respective datasets. It is interesting to note that $R^2$ and RMSE respectively increase and decrease by adding information from different sensors to the model. For example, the predictive power of the model containing multispectral and thermal data is greater than the predictive power of the model constructed with multispectral data





only. The best multiple linear model was the one containing information from all sensors, i.e., multispectral, thermal, and LiDAR data. We therefore used this model (model 1 in Table 5; Fig.5.c) to predict $\Psi_{leaf\_pred}$ at Bousval the 10/08/22, 20/07/23,

10/08/23 and 06/09/23, and at Domaine W the 31/08/22, 20/07/23 and 06/09/23. Due to the technical problems not being able to obtain data from all sensors for certain dates (Table 2), we predicted $\Psi_{leaf\_pred}$ at Bousval the 27/07/22 and 31/08/22 by using the multiple linear model combining multispectral and LiDAR data (model 2 in Table 5; Fig.5.d); we used the multiple linear model combining thermal and LiDAR (model 3 in Table 5; Fig.5.e) to predict $\Psi_{leaf\_pred}$ at Domaine W the 10/08/22. Models 2 and 3 in Table 5 are the most robust and have the best R² and RMSE for the data available at the respective dates and vineyards

(Fig.5.a,b). In each model, the confidence interval on $\Psi_{leaf\_pred}$ increase with decreasing $\Psi_{leaf\_pred}$ (more negative $\Psi_{leaf\_pred}$). The model 1, combining multispectral, thermal and LiDAR data (Fig.5.c), has the lowest uncertainty on the prediction of $\Psi_{leaf\_pred}$ , with a 95 % confidence interval ($CI_{0.95}$) varying between 0.14 MPa (when $\Psi_{leaf\_pred}$ is high) and 0.28 MPa (when $\Psi_{leaf\_pred}$ is low). This is not surprising since this model has the highest predictive power (R²). Model 2 (Fig.5.d), combining multispectral and thermal data, shows a lower uncertainty (0.18 MPa < $CI_{0.95}$ < 0.36 MPa) than the model 3 (Fig.5.e), combining thermal

and LiDAR data (0.20 < $CI_{0.95}$ < 0.40).

**Table 5** – Vegetation index (VI), spectral bands or structural features, and regression weights ($\beta_i$) and intercept ($\alpha$) used in the different multiple linear regression models (equation 3) to predict $\Psi_{leaf}$ ($\Psi_{leaf\_pred}$). $CI_{0.95}$ are 95 % confidence interval on the $\beta_i$

and $\alpha$ of each model; the p-value shows the significance of the variable ($\beta_i$ or $\alpha$) in the model assuming that all other variables exist in the model (significance: p-value < 0.001***; p-value < 0.01**; p-value < 0.05*). Model 1 was constructed based on a combination of multispectral (M), thermal (T) and LiDAR (L) data; Model 2 was constructed based on a combination of multispectral (M) and LiDAR (L) data; Model 3 was constructed based on thermal (T) and LiDAR (L) data. The equations for calculating the VIs contained in each model are given in Table S1.

| | | Vegetation index (VI), spectral bands or structural feature | | | | | | | | Intercept $\alpha$ |
|---|---|---|---|---|---|---|---|---|---|---|
| | | CLRedEdge | CWSIb | Blue | CH | RedEdge | ARI | GNDVI | CST | |
| **Model 1 (M+T+L)** | $\beta_i$ | 0.27 | -0.49 | 8.40 | 0.28 | 0 | 0 | 0 | 0 | -1.55 |
| | $CI_{0.95}$ | [0.18;0.36] | [-0.63;-0.34] | [4.46;12.35] | [0.11;0.44] | 0 | 0 | 0 | 0 | [-1.87;-1.23] |
| | p-value | <0.001*** | <0.001*** | <0.001*** | 0.006** | / | / | / | / | <0.001*** |
| **Model 2 (M+L)** | $\beta_i$ | 0.81 | 0 | 6.78 | 0.38 | 1.49 | 0.02 | -3.32 | 0 | -0.77 |
| | $CI_{0.95}$ | [0.63;0.98] | 0 | [3.19;10.36] | [0.22;0.54] | [0.88;2.11] | [0.01;0.03] | [-4.47;-2.17] | 0 | [-1.43;-0.12] |
| | p-value | <0.001*** | / | 0.04* | <0.001*** | <0.001*** | 0.015* | <0.001*** | / | 0.04* |
| **Model 3 (T+L)** | $\beta_i$ | 0 | -0.28 | 0 | 0.38 | 0 | 0 | 0 | -0.011 | -0.94 |



| | | | | | | | | | |
|---|---|---|---|---|---|---|---|---|---|
| $CI_{0.95}$ | 0 | [-0.51;-0.05] | 0 | [0.22;0.54] | 0 | 0 | 0 | [-0.017;-0.005] | [-1.20;-0.68] |
| p-value | / | 0.04* | / | <0.001*** | / | / | / | 0.002** | <0.001*** |


It is interesting to note that in models 2 and 3 (Table 5), we retrieved the same VIs (CLRedEdge, CWSIb), spectral bands (blue) and structural features (canopy height CH) used to predict $\Psi_{leaf\_pred}$ in the model 1, showing consistency in the parameters used to estimate grapevine $\Psi_{leaf\_pred}$. Multicollinearity between the predictors was controlled by calculating the Variance

Inflation Factor (VIF). For each model, the VIFs were lower than 5 for each VI (Tables S2 to S4), suggesting low multicollinearity between them and enforcing the reliability of each model (James et al., 2021). This absence of multicollinearity between VIs shows that each predictor variable provides significant additional information in the prediction of $\Psi_{leaf\_pred}$. This is confirmed by the partial regression plots (Fig.S6 to S8), showing that each VI used in each model has a significant influence on the prediction of $\Psi_{leaf\_pred}$ (p-value < 0.05). In the model 1, accounting for multispectral, thermal and

LiDAR data, the partial correlations ($R^2$) showed that the CLRedEdge and the CWSIb have the most significant influence on the prediction of $\Psi_{leaf}$ ($R^2 = 0.49$ and $R^2 = 0.56$ respectively), while the blue band ($R^2 = 0.21$) and the canopy height ($R^2 = 0.13$) have less impact (Fig.S5). For the model 2 accounting for the multispectral and LiDAR data, CLRedEdge is also the VI influencing the most the prediction of $\Psi_{leaf\_pred}$ ($R^2 = 0.47$), while the other VIs have less impact ($R^2 < 0.30$) (Fig.S6). CWSIb influenced the most $\Psi_{leaf\_pred}$ ($R^2 = 0.37$) for the model 3 accounting for the thermal and LiDAR data (Fig.S7). Regardless of

the model, the VIs with the greatest influence on $\Psi_{leaf\_pred}$ are therefore the same (CLRedEdge and/or CWSIb). There is therefore a high consistency in the VIs, and in the explanatory power of each VI, for the prediction of $\Psi_{leaf\_pred}$.



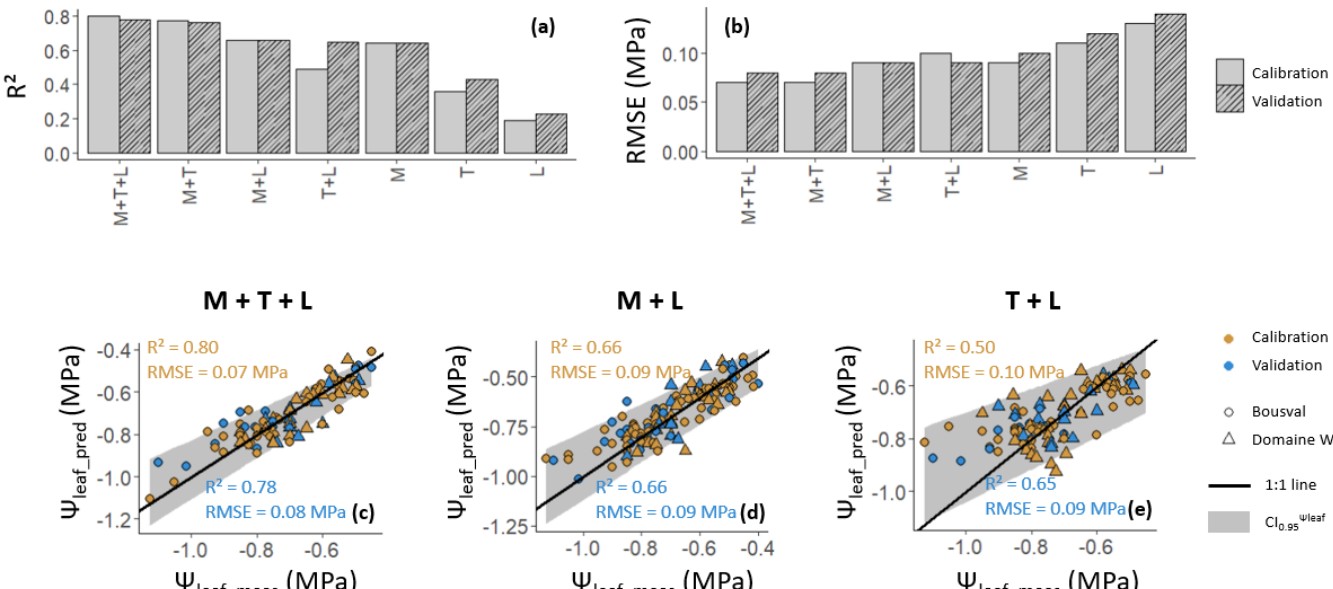


**Figure 5 – (a)** R² and **(b)** RMSE obtained by comparing measured $\Psi_{leaf}$ ($\Psi_{leaf\_meas}$) and predicted $\Psi_{leaf}$ ($\Psi_{leaf\_pred}$) by multiple linear models, for the different data combinations. **(c, d, e)** Relations between $\Psi_{leaf\_meas}$ and $\Psi_{leaf\_pred}$ for the three multiple linear models used in this study. The relationships were fitted with 95% confidence intervals ($CI_{0.95}{}^{\Psi leaf}$ – shaded area). The relation in **(c)** was used to predict $\Psi_{leaf}$ at Bousval the 10/08/22, 20/07/23, 10/08/23 and 06/09/23, and at Domaine W the 31/08/22,

20/07/23 and 06/09/23, by using multispectral (M), thermal (T) and LiDAR (L) data; the relation in **(d)** was used to predict $\Psi_{leaf}$ at Bousval the 27/07/22 and 31/08/22, by using multispectral (M) and LiDAR (L) data; the relation in **(e)** was used to predict $\Psi_{leaf}$ at Domaine W the 10/08/22, by using thermal (T) and LiDAR (L) data.

**3.3 Leaf water potential mapping**


$\Psi_{leaf\_pred}$ maps (Fig.6.a), predicted with multiple regression model obtained with the stepwise regression method, showed a relatively constant pattern over time in both vineyards. At the Bousval vineyard, we observed more negative $\Psi_{leaf\_pred}$ in the eastern part of the plot. In the Domaine W vineyard, $\Psi_{leaf\_pred}$ was lower in the north-western plot than in the south-eastern plot. The spatial heterogeneity of $\Psi_{leaf\_pred}$ is much larger at Bousval than at Domaine W, particularly during the drought

conditions in 2022 (Fig.7.a). For example, the 10/08/22 which was the driest day (Table 1), $\Psi_{leaf\_pred}$ was distributed between -0.52 MPa and -1.25 MPa at Bousval, but between -0.70 MPa and -0.97 MPa at Domaine W. Although the ranges of $\Psi_{leaf\_pred}$ are different, for a same date, the median $\Psi_{leaf\_pred}$ ($\Psi_{leaf\_pred\_median}$) are similar (p-value < 0.05) in both vineyards, excepted for



the 20/07/23 for which $\Psi_{leaf\_pred\_median}$ was slightly greater at Bousval ($\Psi_{leaf\_pred\_median}$ = -0.50 MPa at Bousval; $\Psi_{leaf\_pred\_median}$ = -0.54 MPa at Domaine W). Regarding the temporal dynamics, at Bousval $\Psi_{leaf\_pred}$ decreased between 27/07/22 ($\Psi_{leaf\_pred\_median}$

= -0.74 MPa)  and 10/08/22 ($\Psi_{leaf\_pred\_median}$ = -0.84 MPa), then re-increased the 31/08/22 ($\Psi_{leaf\_pred\_median}$ = -0.76 MPa). We observed similar temporal dynamics at Domaine W, with a re-increase of $\Psi_{leaf\_pred}$ between the 10/08/22 ($\Psi_{leaf\_pred\_median}$ = -0.83 MPa) and 31/08/22 ($\Psi_{leaf\_pred\_median}$ = -0.75 MPa). At Bousval, $\Psi_{leaf\_pred\_median}$ are similar (p-value < 0.05) the 27/07/22 and 10/08/22. In 2023, $\Psi_{leaf\_pred}$ decreased over the season, with $\Psi_{leaf\_pred\_median}$ was -0.50 MPa and -0.54 MPa the 20/07/23 at Bousval and Domaine W respectively, and was -0.74 MPa and -0.73 MPa the 06/09/23 in the same respective vineyards. In

both vineyard we observed similar $\Psi_{leaf\_pred\_median}$ for the last date of 2022 and the last date of 2023 (Fig.7.a).

The uncertainty on $\Psi_{leaf\_pred}$ ($CI_{0.95}^{\Psi leaf\_pred}$), quantified by the 95 % confidence interval on the prediction (Fig.6.b) follows the same spatial pattern than predicted $\Psi_{leaf\_pred}$ (Fig.6.a), with greater uncertainty when $\Psi_{leaf\_pred}$ is more negative. The median value of the uncertainty (median $CI_{0.95}^{\Psi leaf\_pred}$) seems also particularly affected by the model used (Table 5) to estimate $\Psi_{leaf\_pred}$

(Fig.6.c). $\Psi_{leaf\_pred}$ at Domaine W the 10/08/22 shows the greatest uncertainty with a median value of 0.28 MPa. This is not surprising since the model used to predict $\Psi_{leaf\_pred}$ at this date (model 3 in Table 5) only considers thermal and LiDAR data, and is the one showing the lowest predictive power ($R^2$) and the greatest uncertainty (Fig.5.e). For the same date at Bousval, and for a same $\Psi_{leaf\_pred\_median}$ (Fig.7.a), the uncertainty is significantly lower (median $CI_{0.95}^{\Psi leaf\_pred}$ = 0.22 MPa – Fig.6.c), since we used the model involving multispectral, thermal and LiDAR data to estimate $\Psi_{leaf\_pred}$ (model 1 in Table 5), which is the

one showing the highest predictive power ($R^2$) and the lowest uncertainty (Fig.5.c). $\Psi_{leaf\_pred}$ predicted with the model involving multispectral and LiDAR data (model 2 in Table 5) shows the second greater uncertainty, with a median value of 0.26 MPa at Bousval the 27/07/22 and 31/08/22. In both vineyards, the lowest uncertainty was the 20/07/23 (median $CI_{0.95}^{\Psi leaf\_pred}$ = 0.17 MPa), for which we also predicted the highest $\Psi_{leaf}$.

There is a high correlation between $\Psi_{leaf\_pred\_median}$ and the range of $\Psi_{leaf\_pred}$ ($\Psi_{leaf\_pred\_max}$ - $\Psi_{leaf\_pred\_min}$) with a $R^2$ = 0.97 at Bousval, and $R^2$ = 0.94 at Domaine W (Fig.7.b). The range of $\Psi_{leaf\_pred}$ increased when $\Psi_{leaf\_pred\_median}$ decreased. These relations are vineyard-specific (p-value of ANCOVA test is lower than 0.05). The slope of the linear relation between $\Psi_{leaf\_pred\_median}$ and distribution of $\Psi_{leaf\_pred}$ is significantly greater at Bousval (slope = -1.38) than at Domaine W (slope = -0.31).




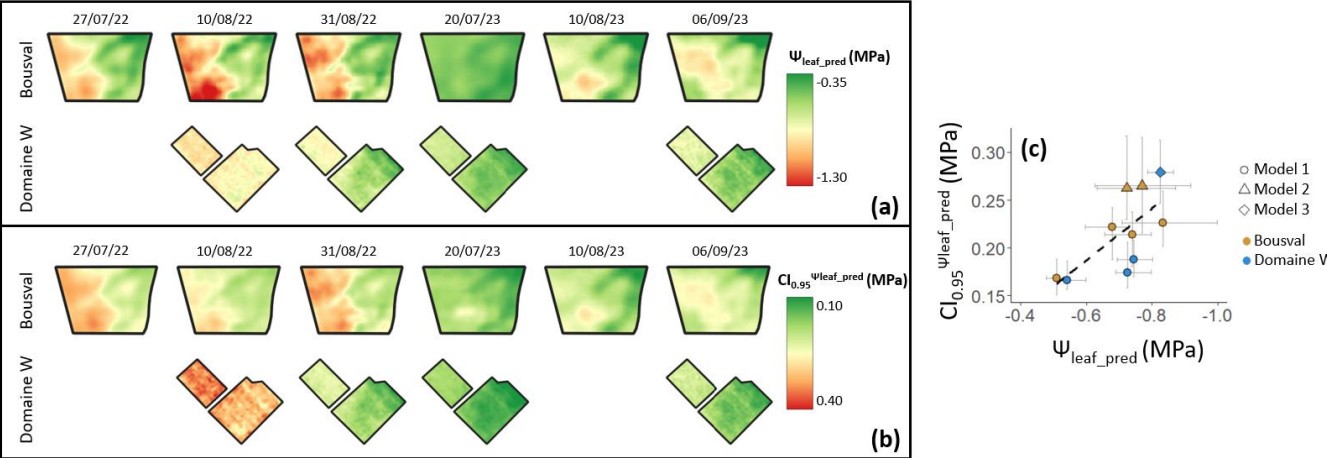

**Figure 6 – (a)** Maps of $\Psi_{leaf}$ predicted ($\Psi_{leaf\_pred}$) with multiple linear regression model in both vineyards. **(b)** Maps of uncertainty on $\Psi_{leaf\_pred}$, quantified by the 95 % confidence interval on the prediction ($CI_{0.95}^{\Psi leaf\_pred}$). $\Psi_{leaf\_pred}$ at Bousval the 10/08/22, 20/07/23, 10/08/23 and 06/09/23, and at Domaine W the 31/08/22, 20/07/23 and 06/09/23 was predicted with the model involving multispectral, thermal and LiDAR data (model 1 in Table 5); $\Psi_{leaf\_pred}$ at Bousval the 27/07/22 and 31/08/22 was predicted with the model involving multispectral and LiDAR data (model 2 in Table 5); $\Psi_{leaf\_pred}$ at Domaine W the 570 10/08/22 was predicted with the model involving thermal and LiDAR data (model 3 in Table 5). **(c)** Relation between $\Psi_{leaf\_pred}$ and $CI_{0.95}^{\Psi leaf\_pred}$. The points correspond to the median, and the horizontal and vertical bars show respectively the quartiles around $\Psi_{leaf\_pred}$ and $CI_{0.95}^{\Psi leaf\_pred}$. The dashed black line is the linear regression; the slope of this regression is different from 0 (p-value < 0.001).





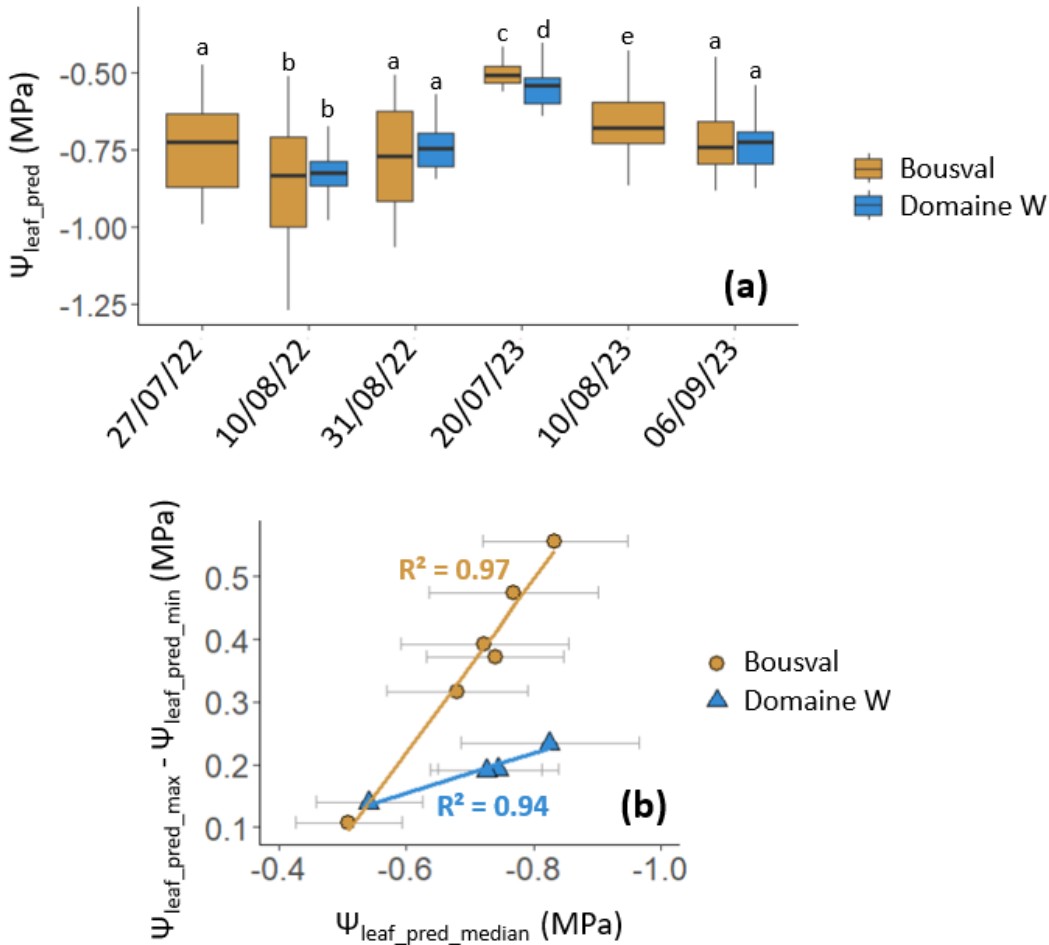

**Figure 7** – **(a)** Boxplots indicating the median and quartiles, minimum and maximum of the spatial distribution of the predicted
$\Psi_{leaf}$ ($\Psi_{leaf\_pred}$). The letters above each boxplot are the results of the Wilcoxon tests; a same letter indicate a statistically similar
(p-value > 0.05) median. **(b)** Relation between the median $\Psi_{leaf\_pred}$ ($\Psi_{leaf\_pred\_median}$) and the distribution of $\Psi_{leaf\_pred}$ ($\Psi_{leaf\_pred\_max}$
- $\Psi_{leaf\_pred\_min}$). The horizontal bars are the median $CI_{0.95}^{\Psi leaf\_pred}$ (median uncertainty) on $\Psi_{leaf\_pred\_median}$.

**3.4 Environmental factors influencing spatial distribution of leaf water potential**

At Bousval, $\Psi_{leaf\_pred}$ was the most correlated with the averaged soil hydraulic conductivity on 2.5 m depth $\widetilde{K}_{soil}$ (Fig.8). The
correlation was better in 2022 than in 2023, with the best correlation the 10/08/22 ($R^2$ = 0.81), the driest day (Table 1). Soil



hydraulic conductivity has therefore a great influence on the spatial distribution of $\Psi_{leaf\_pred}$, particularly when the conditions
are dry. The water holding capacity on 2.5 m depth (WHC) is less correlated with $\Psi_{leaf\_pred}$, however we also find the best
correlation the 10/08/22 ($R^2 = 0.67$). At Bousval, the elevation was moderately correlated with $\Psi_{leaf\_pred}$, particularly in 2022
(e.g. $R^2 = 0.42$ the 10/08/22). This is not surprising since the interface between the loamy and sandy soil horizons is shallower
in the upper part of the parcel (western part) and deeper in the lower part due to an accumulation of loamy colluviums (Fig.1.c).
However, the correlation between $\Psi_{leaf\_pred}$ and elevation is lower than the correlation between $\Psi_{leaf\_pred}$ and $\widetilde{K}_{soil}$, and between
$\Psi_{leaf\_pred}$ and WHC, showing that soil properties have a greater influence on the spatial distribution of $\Psi_{leaf\_pred}$ than the
elevation. The slope showed the lowest correlation with $\Psi_{leaf\_pred}$ (maximum $R^2 = 0.15$ the 31/08/22). It is interesting to note
that the 20/07/23, the correlation between $\Psi_{leaf\_pred}$ and all topographic and soil properties is low (e.g. $R^2 = 0.24$ between
$\Psi_{leaf\_pred}$ and $\widetilde{K}_{soil}$). At Domaine W, the slope and elevation showed a low correlation with $\Psi_{leaf\_pred}$. The maximum $R^2$ between
$\Psi_{leaf\_pred}$ and slope was 0.09, and was 0.24 between $\Psi_{leaf\_pred}$ and elevation both the 20/07/23. These low correlations are not
surprising since this vineyard is almost flat (Fig.1.g).

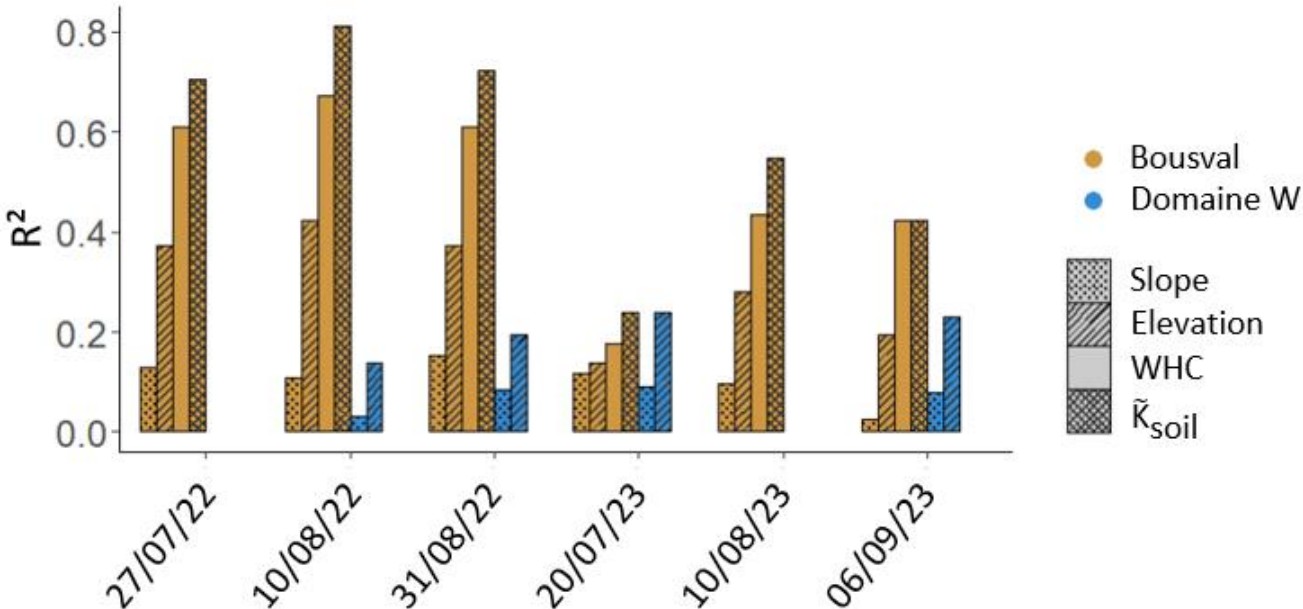

**Figure 8** – Coefficient of determination ($R^2$) of the linear relation between predicted $\Psi_{leaf}$ ($\Psi_{leaf\_pred}$) and elevation, between
$\Psi_{leaf\_pred}$ and slope, between $\Psi_{leaf\_pred}$ and the water holding capacity on 2.5 m (WHC – only at the *Château de Bousval*
vineyard), and between the averaged soil hydraulic conductivity ($\widetilde{K}_{soil}$ – only at the *Château de Bousval* vineyard).






As shown in Fig.7.b, the slope of the linear relation between $\Psi_{leaf\_pred\_median}$ and distribution of $\Psi_{leaf\_pred}$ is significantly greater at Bousval than at Domaine W, showing that environmental conditions at Bousval are much more heterogeneous (i.e. heterogeneity of soil properties), which is reflected in a larger range of distribution of $\Psi_{leaf}$, particularly when conditions are drier. In this study, we quantified the water deficit thanks to the standardized precipitation evapotranspiration index (SPEI –

equation 1). More negative SPEI indicates a greater water deficit. We also used VPD to characterize the atmospheric conditions and quantify the evaporative demand. $\Psi_{leaf\_pred\_median}$ is highly correlated ($R^2$ = 0.82) with the linear combination of SPEI and VPD (Fig.9.a). Interestingly, this relation is not vineyard specific (p-value of ANCOVA test is greater than 0.05). $\Psi_{leaf\_pred\_median}$ is positively correlated with SPEI (regression coefficient a in equation 4 is 0.001) and negatively correlated with VPD (regression coefficient b in equation 4 is -0.13). This means that $\Psi_{leaf\_pred\_median}$ decreases when SPEI decreases and when VPD

increases. In other words, the median $\Psi_{leaf}$ in a vineyard is more negative for greater water deficit (SPEI) and evaporative demand (VPD). The distribution of $\Psi_{leaf\_pred}$ ($\Psi_{leaf\_pred\_max}$ - $\Psi_{leaf\_pred\_min}$) is also correlated ($R^2$ = 0.54) with the linear combination of SPEI and VPD (Fig.9.b). This relation, unlike the one with $\Psi_{leaf\_pred\_median}$, is vineyard-specific (p-value of ANCOVA test is lower than 0.05). The distribution of $\Psi_{leaf\_pred}$ is negatively correlated with SPEI (regression coefficient a in equation 5 is -0.001) and positively correlated with VPD (regression coefficient b in equation 5 is 0.08). This means that the

range of distribution of $\Psi_{leaf\_pred}$ increases when SPEI decreases and VPD increases. In other words, in a vineyard, the spatial heterogeneity of $\Psi_{leaf}$ is more important for greater water deficit (SPEI) and evaporative demand (VPD).

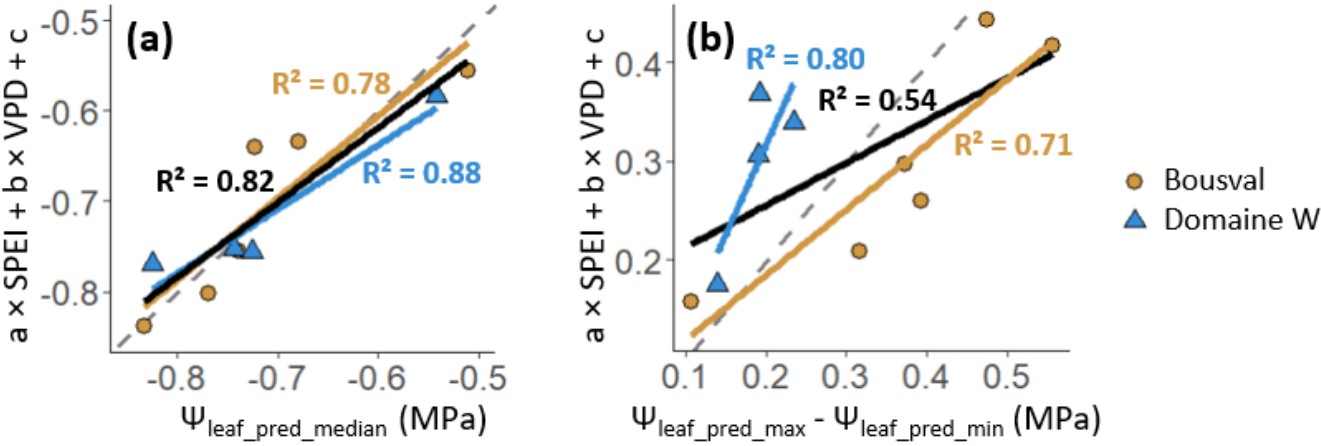

**Figure 9** – **(a)** Relation between median $\Psi_{leaf\_pred}$ ($\Psi_{leaf\_pred\_median}$) and the linear combination of SPEI and VPD (equation 4 – a = 0.001; b = -0.13; c = -0.26). **(b)** Relation between the distribution of $\Psi_{leaf\_pred}$ ($\Psi_{leaf\_pred\_max}$ - $\Psi_{leaf\_pred\_min}$) and the linear





combination of SPEI and VPD (equation 5 – a = -0.001; b = 0.08; c = -0.17). In each subplot, the brown lines are the linear regressions on the brown circles; the blue lines are the linear regressions on the blue triangles; the black lines are the linear regressions on all points; the dashed grey lines are the 1:1 lines.


## 4 Discussion

In this study, we attempted to map grapevine leaf water potential ($\Psi_{leaf}$) intra-field heterogeneity, to assess the impact of topographic, edaphic and climatic conditions on the spatial distribution of $\Psi_{leaf}$. Our study demonstrated that multilinear
combination of multispectral VIs, thermal VIs and structural features from LiDAR data, collected with UAS collecting high-spatial resolution imagery, is efficient ($R^2 = 0.80$ and RMSE = 0.07 MPa for the calibration; $R^2 = 0.78$ and RMSE = 0.08 MPa for the validation) to capture the spatial distribution of grapevine $\Psi_{leaf}$ across different vineyards during two viticultural seasons. We showed that the heterogeneity of edaphic conditions has the greatest influence on the spatial heterogeneity of $\Psi_{leaf}$, particularly when the water deficit and evaporative demand increase.


### 4.1 Leaf water potential prediction with a combination of multispectral, thermal and LiDAR UAS-data

We used the stepwise regression method to find the best multilinear regression to predict $\Psi_{leaf}$ with several UAS-based multispectral VIs, thermal VIs and structural features (LiDAR) measured on grapevines (Wilkinson, 1979). The univariable
linear relation between measured $\Psi_{leaf}$ and VIs (Pearson's ρ > Spearman's ρ – Fig.S3.b) justifies the use of this statistical method to better predict $\Psi_{leaf}$ . However, other statistical methods could also be used to predict $\Psi_{leaf}$, such as principal component regression (PCR) or random forest. To support our approach (stepwise regression), we also implemented the PCR and random forest methods (Fig.S4). However, while the random forest models gave the highest $R^2$ and lowest RMSE for the calibration dataset, there was a great loss of predictive power when we validated the model ($R^2$ decreased and RMSE
increased). This showed some evidence of model overfitting, affecting the robustness of the model constructed with the random forest. Generally, the $R^2$ and RMSE were respectively higher and lower for the stepwise regression model compared to the PCR model. The stepwise regression method showed therefore more robustness than the random forest method when we validated models, and a better predictive power compared to PCR (Fig.S8). Other studies using random forest or artificial neural networks to predict $\Psi_{leaf}$ with UAS-based multispectral VIs lost predictive power when they validated their model,
showing that the model structure was overfitted (Poblete et al., 2017; Romero et al., 2018; Tang et al., 2022). Our analysis highlighted the importance of incorporating multispectral, thermal and LiDAR data to improve the prediction of $\Psi_{leaf}$. The use of data from different sensors also makes it possible to limit the uncertainty on $\Psi_{leaf}$ (Fig.5). Tang et al. (2022) already





mentioned that data combination from multiple sensors acquiring data in different regions of the electromagnetic spectrum will allow mapping $\Psi_{leaf}$ with more accuracy. In our model, the multispectral VI CLRedEdge (calculated based on the NIR

and red-edge bands), the reflectance in the blue band, the thermal VI CWSIb, and the canopy height (CH) all significantly influence $\Psi_{leaf}$ and are used in the multilinear regression to predict it (model 1 in Table 5). It is interesting to note that these indices, bands and features are also used in the models for which only multispectral and LiDAR data (model 2 in Table 5), or thermal and LiDAR data (model 3 in Table 5) are used to predict $\Psi_{leaf}$. Although they are not significantly correlated with measured $\Psi_{leaf}$ when taken one by one (Fig.4), information contained in these indices is complementary and their combinations

enables $\Psi_{leaf}$ to be predicted with high predictive power (e.g. $R^2 = 0.80$ and RMSE = 0.07 MPa for calibration, and $R^2 = 0.78$ and RMSE = 0.08 MPa for calibration, for model 1). It is not required to discard VIs which are not directly correlated with $\Psi_{leaf}$, since multilinear modelling techniques can identify the patterns in the data and assign individual weights to inputs, allowing multilinear models to fit accordingly (Romero et al., 2018). NIR and red-edge spectral regions, used to calculate CLRedEdge, have already been investigated to predict vegetation water potential (Giovos et al., 2021; Pôças et al., 2015;

Soubry et al., 2017; Zygielbaum et al., 2009). Although these bands, and derived VIs, are commonly associated with plant structural traits (e.g., biomass, vigour), red-edge region is often used as a reference to detect chlorophyll content on vegetation (Clevers & Gitelson, 2013; Gamon & Surfus, 1999; Rallo et al., 2014). Plant water status is indicative and closely related to chlorophyll content, as changes in this pigment content induce changes in leaf spectral properties in the red-edge region (Carter & Knapp, 2001). NIR electromagnetic region has also been shown to be affected by leaf structure and leaf water content of

grapevine (De Bei et al., 2011; Marañón et al., 2023; Tardaguila et al., 2017). VIs calculated based on NIR and red-edge bands, such as CLRedEdge, have good potential to quantify grapevine water potential (Becker et al., 2020; Giovos et al., 2021). Additionally, bands in the visible domain, such as the blue band, can provide valuable information about plant water status, as pigment contents and composition govern reflectance in this domain (Gamon et al., 1992; Moya et al., 2004) and are related to processes associated with grapevine water status (Zarco-Tejada et al., 2013). Blue wavelengths are strongly absorbed by

carotenoids (carotenes and xanthophylls). These pigments, and their proportion, also serve as indicators of plant water status (Gitelson et al., 2006). Moreover, the blue reflectance enable atmospheric corrections and allow for a more linear relationship with vegetation status (Gitelson et al., 2002). Thermal VIs, also provide information on grapevine water status. Strong relationships have been found between CWSI and stomatal conductance of grapevine (Pagay & Kidman, 2019; Pou et al., 2014), which is directly linked (non-linearly) to $\Psi_{leaf}$. During drought, grapevines close stomata to prevent the plant from

reaching excessively negative water potentials leading to xylem cavitation (Gambetta et al., 2020). Canopy height (CH), retrieved with LiDAR point clouds, is also used to predict $\Psi_{leaf}$. García-Tejera et al. (2021) showed that changes in plant canopy structure, including canopy height and width, influence the water flow between the soil and the atmosphere, thereby affecting $\Psi_{leaf}$. Thermal remote sensing metrics provide short term information on grapevine water status, such as $\Psi_{leaf}$ or stomatal conductance variations (Acevedo-Opazo et al., 2010; Santesteban et al., 2017), while multispectral VIs and structural features

(LiDAR) reveal mid-to-long-term water status effect on grapevine structure and traits like leaf pigment content (Baluja et al., 2012; Zarco-Tejada et al., 2013). The three approaches (multispectral, thermal and LiDAR) therefore provide complementary





information and their multilinear combinations, through VIs and structural features, allow the accurate and robust assessment of intra-field variability of grapevine leaf water potential, thanks to high-spatial resolution sensors mounted on UAS.

Other sensors could also be tested and used to spatially monitor grapevine $\Psi_{leaf}$. Adding information from more and narrower spectral bands, collected with hyperspectral sensors, could improve the capability to remotely monitor grapevine $\Psi_{leaf}$ (Pôças et al., 2015; Tang et al., 2022). Studies using hyperspectral sensors to measure plant water potential generally get higher correlations than studies using multispectral sensors (Pôças et al., 2015; Rodríguez-Pérez et al., 2007; Zarco-Tejada et al., 2013). For example, the photochemical reflectance index (PRI, calculated with spectral reflectance at 545 nm and 567 nm) is

a good indicator ($R^2$ between 0.5 and 0.6) of crop water status (Stagakis et al., 2012; Suárez et al., 2008). Zarco-Tejada et al. (2013) even obtained a better correlation ($R^2 = 0.82$) in vineyards by combining PRI with other hyperspectral VIs, such as the renormalized difference vegetation index (RDVI, based on reflectance at 700 nm and 761 nm) and the ratio $R_{700}/R_{670}$ (based on reflectance at 670 nm and 700 nm), highlighting that the combination of several VIs brings complementary information to better estimate leaf water potential of grapevine. Short-wave infrared (SWIR) data (1000-2200 nm) also has a good potential

to monitor grapevine water status, since this spectral range contains the water absorption bands (Laroche-Pinel, Albughdadi, et al., 2021). VIs calculated with SWIR bands, such as the normalized drought water index (NDWI) (Gao, 1996), showed good correlations ($R^2 = 0.58$) with grapevine $\Psi_{leaf}$ (Caruso & Palai, 2023).

### 4.2 Intra-field variability of grapevine leaf water potential


We observed a good stability in $\Psi_{leaf}$ pattern for each date, in both vineyards (Fig.6.a). At the Domaine W vineyard, the spatial heterogeneity of $\Psi_{leaf}$ is less marked than at the Bousval vineyard. For example, the 10/08/22 (driest day), although median $\Psi_{leaf}$ was similar in both vineyards, range of distribution of $\Psi_{leaf}$ was 0.73 MPa at Bousval, but only 0.27 MPa at Domaine W. The magnitude of variation $\Psi_{leaf}$ at the within field level predicted at Bousval is consistent with other studies. Brillante et al.

(2017a) also observed a difference of 0.70 MPa between maximum and minimum $\Psi_{leaf}$ within a vineyard in California. They hypothesized, without directly proving it, that this variability was due to the short distance differences of soil properties in the vineyard. Tang et al. (2022) observed a spatial variability up to 0.67 MPa, but this was due to differences in irrigation treatment in a vineyard with gravelly loam soil. At Bousval, the spatial distribution of $\Psi_{leaf}$ is highly correlated (Fig.8 – $R^2$ up to 0.81) with an averaged soil hydraulic conductivity $\widetilde{K}_{soil}$. In this vineyard, $\Psi_{leaf}$ was significantly more negative in the eastern part of

the plot, where the interface between the loamy soil and the sandy subsoil is more superficial, compared to the western part where the loamy soil is clearly deeper. Grapevine water potential is significantly influenced by the soil texture. It is well known that the soil water potential around the roots affects $\Psi_{leaf}$, since the difference between both water potentials is the driving force for transpiration (Tyree & Zimmermann, 2002). The soil water potential in the vicinity of the roots decreases as the plant takes up water, resulting in a significant loss of soil hydraulic conductivity around the roots (Cai et al., 2022). This reduction of





hydraulic conductivity generates large gradients in soil water potential at the vicinity of the roots, leading to a significant drop

of $\Psi_{leaf}$ to support a slight increase in transpiration (Carminati & Javaux, 2020). The soil texture determines soil hydraulic

properties, thereby influencing grapevine hydraulics (Lavoie-Lamoureux et al., 2017; Tramontini et al., 2013). Therefore, in a

sandy soil, the decline in soil hydraulic conductivity around the roots is sharper than in a fine-textured soil, leading to a

significantly greater reduction of water potential at the soil-root interface, which directly impacts $\Psi_{leaf}$ which also decreases

more rapidly (Cai et al., 2022). In soil-water limited conditions, such as in 2022 (Fig.S1), we can assume that soil hydraulic

conductivity drop is larger in the western part of the Bousval vineyard, where most of the grapevine roots are found in the

sandy subsoil (Delval et al., submitted), leading to a greater drop of soil-root interface water potential and consequently to a

larger decline of $\Psi_{leaf}$. We can therefore affirm that in a non-irrigated vineyard, the edaphic heterogeneity (i.e. in terms of soil

hydraulic conductivity) governs the spatial heterogeneity (and patterns) of grapevine $\Psi_{leaf}$, particularly during drought. At the

Domaine W vineyard, we always predicted lower $\Psi_{leaf}$ in the north-west parcel than in the south-east plot. We observed, but

only in a single location in each subplot, that the north-west parcel is made of a silty loam soil on the first horizons and silty

clay loam soil thereafter, while the south-east parcel is composed by a silty loam soil on the whole profile (Fig.1.h). Although

the vertical distribution of soil texture is not accurately known in the whole field, we could assume that this spatial difference

of soil properties has an influence on the spatial heterogeneity of $\Psi_{leaf}$, since other factors such as the elevation ($R^2 = 0.24$) and

the slope ($R^2 = 0.09$) have low correlation with the spatial distribution of $\Psi_{leaf}$ (Fig.8). However, other studies comparing loamy

and loamy clay soils showed only a moderate effect on grapevine $\Psi_{leaf}$ (Brillante, Mathieu, et al., 2017), explaining the lower

range of $\Psi_{leaf}$ observed in this vineyard. Other factors could potentially influence the spatial distribution of $\Psi_{leaf}$ at the Domaine

W. For example, in this vineyard, the level of the water table is higher in the south-eastern parcel, due to the stream running

parallel to the plot (Fig.S1.c). It has been observed that this water table reach the roots, and water consumed by the grapevines

is therefore certainly replaced by vertical soil-water movements (capillary rises) (Van Leeuwen et al., 2018). The presence of

a water table within the reach of the roots prevent or mitigate decrease of $\Psi_{leaf}$ (Tramontini et al., 2013). Further investigations

should be done to better understand the spatial heterogeneity of $\Psi_{leaf}$ in the Domaine W vineyard.

We showed that the linear combination of SPEI and VPD was correlated with the median ($\Psi_{leaf\_median} - R^2 = 0.82$) and the

range ($\Psi_{leaf\_max} - \Psi_{leaf\_min} - R^2 = 0.54$) of leaf water potential (Fig.9). In a vineyard, the $\Psi_{leaf\_median}$ is more negative when the

water deficit and the evaporative demand is greater (i.e. respectively when SPEI is lower and VPD is higher). Interestingly,

we showed that this relation was independent of the vineyard (p-value > 0.05). The range of $\Psi_{leaf}$ is more important for greater

water deficit and evaporative demand. This relation was vineyard-dependent (p-value < 0.05). While the edaphic heterogeneity

can explain the $\Psi_{leaf}$ spatial heterogeneity observed within a vineyard, the median and range of $\Psi_{leaf}$ are particularly affected

by the weather conditions (i.e. evaporative demand) and the intensity of water deficit. When evaporative demand and/or water

deficit are greater, the spatial heterogeneity of $\Psi_{leaf}$ is particularly marked, and follows $\widetilde{K}_{soil}$ intra-field patterns. Therefore,

weather conditions also have a great influence on the temporal variability of $\Psi_{leaf}$ (Brillante, Martínez-Luscher, et al., 2017).

The impact of weather conditions on $\Psi_{leaf}$ can explain why we observed a re-increase of $\Psi_{leaf\_median}$ between the 10/08/22 and



31/08/22 in both vineyards, since VPD on 10/08/22 was significantly higher than VPD on 31/08/22. Plants exposed to a higher
evaporative demand experience a greater loss in water potential at the soil-root interface, resulting in more negative $\Psi_{leaf}$.
Conversely, for grapevines exposed to lower VPD, the drop in $\Psi_{leaf}$ is more limited since the water potential at the soil-root
interface is higher (Cai et al., 2022; Carminati & Javaux, 2020). Spatial soil properties distribution, weather conditions and
intensity of water deficit mainly influence grapevine leaf water potential heterogeneity, and the median and range of $\Psi_{leaf}$
observed in a vineyard, and their effects are concomitants (Van Leeuwen et al., 2018).


It is interesting to note that the 20/07/23, $\Psi_{leaf}$ is relatively homogeneous within both vineyards and its distribution ($\Psi_{leaf\_max}$ -
$\Psi_{leaf\_min}$) is therefore low (Fig.7.a). At Bousval, $\Psi_{leaf}$ was still slightly lower in the western part of the parcel, but the difference
was significantly less marked compared to the other dates (Fig.6). Same observations can be made at Domaine W, with slightly
more negative $\Psi_{leaf}$ in the north-western plot, but a less marked variability compared to other dates. The 20/07/23, the soil was
the wettest ever measured for this study (Fig.S1), as well as the water deficit (SPEI) and the evaporative demand was the
lowest among all dates (Table 1). In non-limiting soil conditions, water flow is primarily governed by plant hydraulic
conductance, instead of soil hydraulic conductivity, even in sandy soils (Draye et al., 2010; Passioura, 1980). Therefore, plant
hydraulic conductance mainly affects leaf water potential distribution in these conditions. Although it is well-known that
edaphic conditions influence grapevine hydraulic conductance (Tramontini et al., 2013), notably through their impact on xylem
(Hochberg et al., 2015), root (Ollat et al., 2015) and canopy (Pereyra et al., 2023) architecture, we can assume that the within-
field grapevine hydraulic conductance is significantly less heterogenous than within-field soil hydraulic conductivity. This is
not surprising since in this study, for a given vineyard, we only worked on one cultivar-rootstock combination. Although some
studies highlighted the predominance of the soil effect on grapevine water potential (Taylor et al., 2010; Tramontini et al.,
2013), it would be interesting to carry out research to understand how, within a same vineyard with different cultivar-rootstock
combinations, this affects the range of within-field $\Psi_{leaf}$.

Other studies showed that topographic attributes, such as slope and elevation, could also impact grapevine performance
(Bramley et al., 2011; Karn et al., 2024). In this study, we only observed a maximum $R^2 = 0.13$ and $R^2 = 0.54$ respectively
between $\Psi_{leaf}$ and slope, and between $\Psi_{leaf}$ and elevation. This is consistent with Brillante et al. (2017a) who showed that slope
and elevation differences are less significantly related to grapevine water status heterogeneity in vineyards with moderate or
no slope, which is the case in the present study. It has been shown that topographic attributes have a real influence on grapevines
for vineyards with steep slopes (Brillante, Mathieu, et al., 2017).



## 5 Conclusion

We aimed to accurately map the grapevine leaf water potential ($\Psi_{leaf}$) within non-irrigated vineyards and assess the impact of edaphic, topographic and climatic conditions on the $\Psi_{leaf}$ intra-field heterogeneity. We combined UAS-based multispectral, thermal and LiDAR data to spatially predict grapevine $\Psi_{leaf}$  The data provided by different sensors acquiring data in different regions of the electromagnetic spectrum brought complementary information on grapevine water status and allowed the development of a robust and high-predictive power model ($R^2 = 0.80$ and RMSE = 0.07 MPa for the calibration; $R^2 = 0.78$ and RMSE = 0.08 MPa for the validation) to estimate grapevine $\Psi_{leaf}$ in two vineyards, during two viticultural seasons. While thermal VIs (e.g. CWSI) provide short term information, such as $\Psi_{leaf}$ variations, multispectral (e.g. CLRedEdge and blue reflectance band) and LiDAR (from which we can derived grapevine structural features) data are associated to mid-to-long term water status effect on grapevine structure (e.g. canopy height) and traits like leaf pigment content. Our results provided evidence that in non-irrigated vineyards, grapevine water status is highly variable within a vineyard, up to 0.73 MPa. This spatial distribution of $\Psi_{leaf}$ is mainly governed by the within-vineyard soil hydraulic conductivity heterogeneity, and is particularly marked when the evaporative demand and the water deficit are greater, since the range of $\Psi_{leaf}$ increases in these conditions. Knowledge of spatial variability of grapevine water status, through grapevine $\Psi_{leaf}$, could help winegrowers to accurately optimize the viticultural management during the different phenological stages of the grapevine. Although promising, our results are limited to one grapevine cultivar (cv. Chardonnay) and two vineyards. To further improve the robustness and reliability of the method used in this study, additional UAS observations should be done to represent a broader range of cultivars, rootstocks, management systems and environmental conditions, to examine how other viticultural factors may affect grapevine $\Psi_{leaf}$ spatial heterogeneity. Moreover, accurate spatialized information of grapevine $\Psi_{leaf}$ could be used in functional-structural grapevine models (Yang et al., 2023) to predict berry growth and quality (i.e. sugar content) and its variation at the field scale.



**Data availability**

The data collected and/or analysed during the current study are available from the corresponding author upon reasonable request.


**Author contributions**

LD, FJ and MJ conceived the study. LD, FJ and MJ designed the experiments. LD and JB performed the experiments, LD analysed the data. LD wrote the manuscript with input from all authors.

**Competing interests**

The authors declare that they have no conflict of interest.

**Acknowledgements**

We thank the managers of *Château de Bousval* and *Domaine W* vineyards for allowing us to collect data in their vineyards, in complete freedom and transparency. This work was supported by a FRIA grant from the Belgian Fund for Scientific Research FSR-FNRS [grant FC041167].



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
