# Peer review of "Field heterogeneity of soil texture controls leaf water potential spatial distribution predicted from UAS-based vegetation indices in non-irrigated vineyards"

_EGUsphere, 2024_

## Author Response (AR1)

**Earth and Life Institute**
**Department of Environmental Sciences**

UCLouvain
Louis Delval
louis.delval@uclouvain.be

**Response to reviewers**

Dear Editor,

Thank you very much for your decision regarding our manuscript "Field heterogeneity of soil texture controls leaf water potential spatial distribution predicted from UAS-based vegetation indices in non-irrigated vineyards" (the title has been modified based on the reviewer suggestions). We have carefully read both the reviewer's suggestions and revised the manuscript accordingly. Below you can find our point-to-point answers to the reviewers.

Yours sincerely,

Louis Delval, Jordan Bates, François Jonard and Mathieu Javaux

**Reviewer #1 – Prof. Cornelis van Leuween**

*Answers in blue
**Line numbers refer to the version "Track-Changes"

We would like to thank Prof. Cornelis Van Leeuwen for the positive review and comments of our preprint. We particularly appreciate the emphasis on the originality and limitations of our study. The minor modifications suggested by Prof. Cornelis Van Leeuwen have been taken into account to improve the paper.

I suggest the title to be modified as « Field heterogeneity of soil texture controls leaf water potential spatial distribution predicted from vegetation indexes in non-irrigated vineyards »

We modified the title as "Field heterogeneity of soil texture controls leaf water potential spatial distribution predicted from UAS-based vegetation indices in non-irrigated vineyards"

P5, line 120   Delete « the interface between »

Line 131 → It has been deleted.

P13, line 366  Delete « and »

Line 380 → It has been deleted.

P17, line 455  In Figure 4, replace « CI » by « CWSIb »

CI and CWSIb are two different vegetation indices. Both are present in figure 4.

P17, line 460  Add « s » to model (« multiple linear regression models »)

Line 477 → It has been added.

P20, line 528  Replace « eastern » by « western »

Line 548 → It has been replaced.

P26, lines 651 and 653  Two following sentences start with « However », please reword.

Line 690 → It has been reworded.

P42, line 1077 Replace « OENO One » by *Journal International des Sciences de la Vigne et du Vin* (all articles initially published in the *J. Int. Sci. Vigne Vin* are available on the OENO One website, but in 2009 OENO One did not yet exist).

Line 1166 → It has been replaced.

**Reviewer #2 – Dr. Clément Saint-Cast**

We would like to thank Dr. Clément Saint-Cast for the interesting review and comments of our preprint. All the suggestions will help us improve the quality of our article.

The authors combined multilinearly vegetation indices from multispectral, thermal and LiDAR sensors to capture the spatial distribution of grapevine water status within vineyards. To discriminate information from the vine canopy pixels to inter-row soil, they developed an interesting analytical pipeline, integrating K-means clustering analysis and different vegetation indices. I'm not familiar with this procedure and whether it has been done in previous studies. If this new pipeline has never been published, I encourage the authors to write a short paragraph on this topic in the discussion, detailing the added value of developing this approach compared to what has been done before. It should again highlight the novelty and interest of this study. Moreover, it should support the advantages of using several vegetation indices from different sensors: performing a better discrimination of the vine pixel and performing a spatial distribution of the vine water status.

We added, in the Discussion, a paragraph (section 4.1) explaining the originality and added value of our methodology, and discuss and compare the different methods to be found in the literature.

The authors mentioned a high spatial resolution of the vine water status captured by their UAS devices. However, I wondered how important this resolution is compared to what has been done in the past. In fact, I did not see any reference and comparison with previous studies. In addition, it would be interesting to know the size of the pixel of the map obtained at the end of the "grapevine pixel extraction". These values could be mentioned in the abstract or in the conclusion, and perhaps compared with the values of previous studies in the discussion, in order to highlight the quality of this work. In the same way, it would be good to add scale bars with the different maps in Figure 2.

We now better mention the spatial resolution of our final maps in the article, and compare it to values in other recent studies. We also added scale bars in the different maps in Figure 2.

The authors efficiently capture the spatial distribution of $\Psi$leaf by testing different combinations of vegetation indices using a stepwise regression method. The results are consistent and the authors find that of the seven data combinations tested, the vegetation indices with the strongest predictive ability are CLRedEdge, NDRE and CWSIb. The authors show a better predictive power of the models with LiDAR data. I'm a bit surprised by this result and wonder what additional information the LiDAR provides to better capture the water status of the vine in this experiment? Indeed, "structural" responses of grapevine to water deficit are often observed at low water potentials (e.g. $\Psi$ stem < -1.6 MPa induces leaf shedding). The lowest water potential recorded was -1.15 MPa (Table 4), corresponding to a moderate water deficit. So it's surprising to observe an added value from the use of LiDAR, and I wonder how you explain the contribution of the LiDAR sensors to the model? It's lower growth induced by stomatal closure?

Perhaps you could add a sentence to the discussion explaining the biological process that might explain this result. I also noticed a low contribution of the LiDAR to the $R^2$ and RMSE in Figures 5a and 5b (comparison between M+T+L and M+T; M+L and M).

We now better explain the biological and physiological processes that might explain and justify the added value of LiDAR data to predict leaf water potential.

The study presented here focuses on two vineyards in Belgium. The reason for using these two different vineyards is not very well explained and could be better introduced in the manuscript.

We now better justify why we chose those vineyards in the Methodology.

In the results part, it might be interesting to provide the values of the results highlighted by the authors.

We now provided more values everywhere in the manuscript.

Please consider if "weather" (e.g. line 11 and 18) is the best term here. Perhaps "climatic conditions" is more appropriate.

Lines 13 and 20 → We replaced "weather" by "climatic conditions"

P.8 L. 204: Add a space between "$m^2$" and "zones".

Line 215 → It has been added.

P11. L.294-296: Perhaps give the reason for focusing on these $2x2m^2$ zones.

Line 309 → We now give the reason to focus on these $2x2\ m^2$ zones is to predict leaf water potential.

P16. L.441: Delete the point after "$\Psi leaf\_meas$".

Line 458 → It has been deleted.

P16. 444: I suggest deleting "It is interesting to note". Let the reader decide if it's interesting or not. So just be factual and present the results without taking a position: "Pearson is greater than Spearman". Same comment on line 496 or for similar formulations (e.g. "This is not surprising to").

Lines 461 and 513 → It has been deleted.

P.20 L.528: Replace "eastern part" by "western part" (the same comment for the line 724).

Lines 544 and 766 → It has been replaced.

Fig. 6: For ease of reading, the name of the model used to reconstruct the map could be added for each date (e.g. 27/07/22 - Model 1 or 27/07/22 – M.1).

It has been added on Figure 6.

P.24 L.589-590: I suggest adding this sentence to the discussion.

Line 605 → The sentence has been moved in the discussion.

P.27 L.691: Put a dot after "Ψleaf". Same on line 799.

It has been added.